# Blind spots in global soil biodiversity and ecosystem function research

Carlos A. Guerra [1,2 ✉], Anna Heintz-Buschart [1,3], Johannes Sikorski[4], Antonis Chatzinotas [5,1], Nathaly Guerrero-Ramírez[1,6], Simone Cesarz[1,6], Léa Beaumelle [1,6], Matthias C. Rillig [7,8], Fernando T. Maestre [9,10], Manuel Delgado-Baquerizo [9], François Buscot [3,1], Jörg Overmann [4,11], Guillaume Patoine[1,6], Helen R. P. Phillips [1,6], Marten Winter [1,6], Tesfaye Wubet [12,1], Kirsten Küsel [1,13], Richard D. Bardgett [14], Erin K. Cameron[15], Don Cowan[16], Tine Grebenc [17], César Marín [18,19], Alberto Orgiazzi [20], Brajesh K. Singh [21,22], Diana H. Wall [23] & Nico Eisenhauer [1,6]

Soils harbor a substantial fraction of the world's biodiversity, contributing to many crucial ecosystem functions. It is thus essential to identify general macroecological patterns related to the distribution and functioning of soil organisms to support their conservation and consideration by governance. These macroecological analyses need to represent the diversity of environmental conditions that can be found worldwide. Here we identify and characterize existing environmental gaps in soil taxa and ecosystem functioning data across soil macroecological studies and 17,186 sampling sites across the globe. These data gaps include important spatial, environmental, taxonomic, and functional gaps, and an almost complete absence of temporally explicit data. We also identify the limitations of soil macroecological studies to explore general patterns in soil biodiversity-ecosystem functioning relationships, with only 0.3% of all sampling sites having both information about biodiversity and function, although with different taxonomic groups and functions at each site. Based on this information, we provide clear priorities to support and expand soil macroecological research.

[1] German Centre for Integrative Biodiversity Research (iDiv) Halle-Jena-Leipzig, Leipzig, Germany. [2] Institute of Biology, Martin Luther University Halle Wittenberg, Am Kirchtor 1, 06108 Halle(Saale), Germany. [3] Helmholtz Centre for Environmental Research - UFZ, Department of Soil Ecology, 06108 Halle (Saale), Germany. [4] Leibniz-Institut DSMZ-Deutsche Sammlung von Mikroorganismen und Zellkulturen, Braunschweig, Germany. [5] Helmholtz Centre for Environmental Research - UFZ, Department of Environmental Microbiology, Leipzig, Germany. [6] Institute of Biology, Leipzig University, Leipzig, Germany. [7] Freie Universität Berlin, Institut für Biologie, Altensteinstr. 6, 14195 Berlin, Germany. [8] Berlin-Brandenburg Institute of Advanced Biodiversity Research (BBIB), Altensteinstr. 34, 14195 Berlin, Germany. [9] Departamento de Biología y Geología, Física y Química Inorgánica, Escuela Superior de Ciencias Experimentales y Tecnología, Universidad Rey Juan Carlos, Calle Tulipán Sin Número, Móstoles 28933, Spain. [10] Departamento de Ecología and Instituto Multidisciplinar para el Estudio del Medio "Ramón Margalef, Universidad de Alicante, Carretera de San Vicente del Raspeig s/n, 03690 San Vicente del Raspeig, Alicante, Spain. [11] Microbiology, Braunschweig University of Technology, Braunschweig, Germany. [12] Helmholtz Centre for Environmental Research - UFZ, Department of Community Ecology, Braunschweig, Germany. [13] Institute of Biodiversity, Friedrich Schiller University Jena, Dornburger-Straße 159, 07743 Jena, Germany. [14] School of Earth and Environmental Sciences, The University of Manchester, Manchester M13 9PT, UK. [15] Department of Environmental Science, Saint Mary's University, Halifax, NS, Canada. [16] Centre for Microbial Ecology and Genomics, Department of Biochemistry, Genetics and Microbiology, University of Pretoria, Pretoria, South Africa. [17] Slovenian Forestry Institute, Večna pot 2, SI-1000 Ljubljana, Slovenia. [18] Instituto de Ciencias Agronómicas y Veterinarias, Universidad de O'Higgins, Rancagua, Chile. [19] Instituto de Ciencias Ambientales y Evolutivas, Universidad Austral de Chile, Valdivia, Chile. [20] European Commission, Joint Research Centre (JRC), Ispra, Italy. [21] Hawkesbury Institute for the environment, Western Sydney University, Penrith, NSW 2751, Australia. [22] Global Centre for Land-Based Innovation, Western Sydney University, Penrith, NSW 2751, Australia. [23] School of Global Environmental Sustainability and Department of Biology, Colorado State University, Fort Collins, CO 80523-1036, USA. ✉email: carlos.guerra@idiv.de

Soils harbor a large portion of global biodiversity, including microorganisms (e.g., bacteria), micro- (e.g., Nematoda), meso- (e.g., Collembola), and macrofauna (e.g., Oligochaeta)[1]. This high biodiversity plays critical roles driving multiple ecosystem functions and services, including climate regulation, nutrient cycling, and food production[1–6]. Accordingly, recent experimental[7,8] and observational[9,10] studies, based either on particular biomes (e.g., drylands) or local sites, have shown that soil biodiversity is of high importance for the maintenance of multifunctionality (i.e. the ability of ecosystems to simultaneously provide multiple ecosystem functions and services[11]) in terrestrial ecosystems.

Nevertheless, and with few exceptions[9,12], global soil biodiversity-ecosystem function relationships have not yet been studied in depth in macroecological perspectives and evaluations of patterns and causal mechanisms that link soil biodiversity to soil ecosystem functions have only emerged in the last decade[10,13–15]. By comparison, albeit with important limitations[16], there is a plethora of studies describing the global distribution and temporal patterns of aboveground biodiversity[17], ecosystems[18], and biodiversity-ecosystem function relationships[12,19–23], something that is currently mostly absent (but see Delgado-Baquerizo et al.[24]) in soil macroecological studies due to the lack of temporally explicit data for soil biodiversity and soil-related functions.

Despite the mounting number of soil ecology studies, major gaps and/or geographic and taxonomic biases exist in our understanding of soil biodiversity[25]. Although the existing gaps in global soil biodiversity data are consistent with gaps in other aboveground biota[16,26,27], these are further exacerbated when described across specific ecological gradients (e.g., differences across altitudinal gradients) and taxa (e.g., Collembola, Oligochaeta)[28]. Furthermore, almost nothing is known about temporal patterns in soil biodiversity at larger spatial scales and across ecosystem types[25]. Identifying and filling these gaps on soil taxa distributions and functions is pivotal to identify the ecological preferences of multiple soil taxa, assess their vulnerabilities to global change, and understand the causal links between soil biodiversity, ecosystem functioning, and associated ecosystem services[16,29]. Despite growing scientific and political interest in soil biodiversity research[25], little to no attention is given to the governance of soil ecosystems (Supplementary Fig. 1). This has resulted in a lack of inclusion of soil biodiversity and functions in land management and conservation debates, and environmental policy[30].

In contrast to groups of organisms from other ecosystems (e.g., aboveground terrestrial[31]) for which the Global Biodiversity Information Facility (GBIF) constitutes already the main global data hub[32,33], soil organisms are poorly represented. In fact, distribution data on soil taxa are spread across the literature, museum archives, and a number of non-interoperable platforms (e.g., EDAPHOBASE (https://portal.edaphobase.org/), the global Ants database[34], the Earth Microbiome project[35]), and much needs to be done to fully aggregate these valuable resources. Across all available soil biodiversity data, major issues remain regarding their spatial and temporal representativeness (e.g., absent data in most tropical systems), and coverage of taxonomic groups (e.g., focus on fungi and bacteria), which limits our capacity to comprehensively assess and understand soil systems at multiple temporal and biogeographic scales. Also, even for the most represented taxa (i.e., bacteria and fungi), there are strong concerns regarding the current taxonomic depth[36], even in better covered regions.

More importantly, both the lack of representativeness and the distribution of gaps in global soil biodiversity and ecosystem function research hampers the prioritization of future monitoring efforts[16]. Such knowledge deficit in soil biodiversity also prevents stakeholders from taking appropriate management actions to preserve and maintain important ecosystem services[37], such as food and water security, for which soils are the main provider[1]. Therefore, it is both timely and relevant to identify these blind spots in global soil macroecological knowledge and research. By doing so, we can assess their main causes and line up potential solutions to overcome them.

Since the mere accumulation of data will not advance ecological understanding[38,39], it is important to identify how well the current macroecological studies cover the range of existing environmental conditions on Earth, including soil properties, climate, topography, and land cover characteristics[40,41]. Therefore, here we identify fundamental gaps in soil macroecological research by analyzing the distribution of sampling sites across a large range of soil organisms and ecosystem functions. In a review of current literature, we collected sample locations from most existing studies focused on soil macroecological patterns (see below; Table 2). The studies were then organized according to different soil taxonomic groups and ecosystem functions studied (nine and five categories, respectively, see Methods for more details). Finally, we examined how these macroecological studies have captured the diversity of global environmental conditions to identify critical ecological and geographical "blind spots" of global soil ecosystem research (e.g., specific land use types, soil properties, climate ranges; see Methods for more detail). By identifying the environmental conditions that have to be covered in future research and monitoring to draw an unbiased picture of the current state of global soils as well as to reliably forecast their futures, our synthesis goes a step beyond recent calls to close global data gaps[25]. Our comprehensive spatial analysis will help researchers to design future soil biodiversity and ecosystem function surveys, to support the mobilization of existing data, and to inform funding bodies about the allocation of research priorities in this important scientific field.

## Results and discussion

**Biogeographical biases.** From our literature search, we collected details on locations of 17,186 individual locations/sampling sites representing macroecological studies on soil biodiversity ($N = 12,915$) and ecosystem functions ($N = 3318$) (Fig. 1). In our assessment, we also included studies on soil organisms referring to organism biomass ($N = 977$; e.g. microbial and faunal biomass) as an important link between biodiversity and function, although our focus will be on the last two components. Bacteria, fungi, and soil respiration (Fig. 1a) were the best-represented soil taxa and functions in our literature survey, respectively. The total number of sites across all studies is low compared with many aboveground macroecological databases that surpass the numbers found here (e.g., the PREDICTS database[42] contains ~29,000 sites across the globe).

Globally, soil biodiversity and ecosystem function data are not evenly distributed. Bacteria ($N = 3453$), fungi ($N = 1687$), and Formicoidea ($N = 3024$; which together concentrate 48.8% of all soil biodiversity records) have comparatively large and geographically balanced distributions when compared to Rotifera ($N = 41$), Collembola ($N = 27$), and Acari ($N = 10$), which have a substantially lower number of sampling sites and more scattered distributions (see Supplementary Fig. 2 for more detail). The distribution and availability of data for macroecological studies of soil organisms has changed dramatically during the last year, with the outcomes of efforts to synthesize local scale studies into large-scale initiatives[43,44]. Thus, in the case of bacteria, fungi, Nematoda, and Oligochaeta (here including earthworms and enchytraeids), the relatively high number of sampling sites reflects a community effort to assemble databases based on collections from different projects[10,45]. In the case of Formicoidea, the availability of data reflects the outcome of systematic global

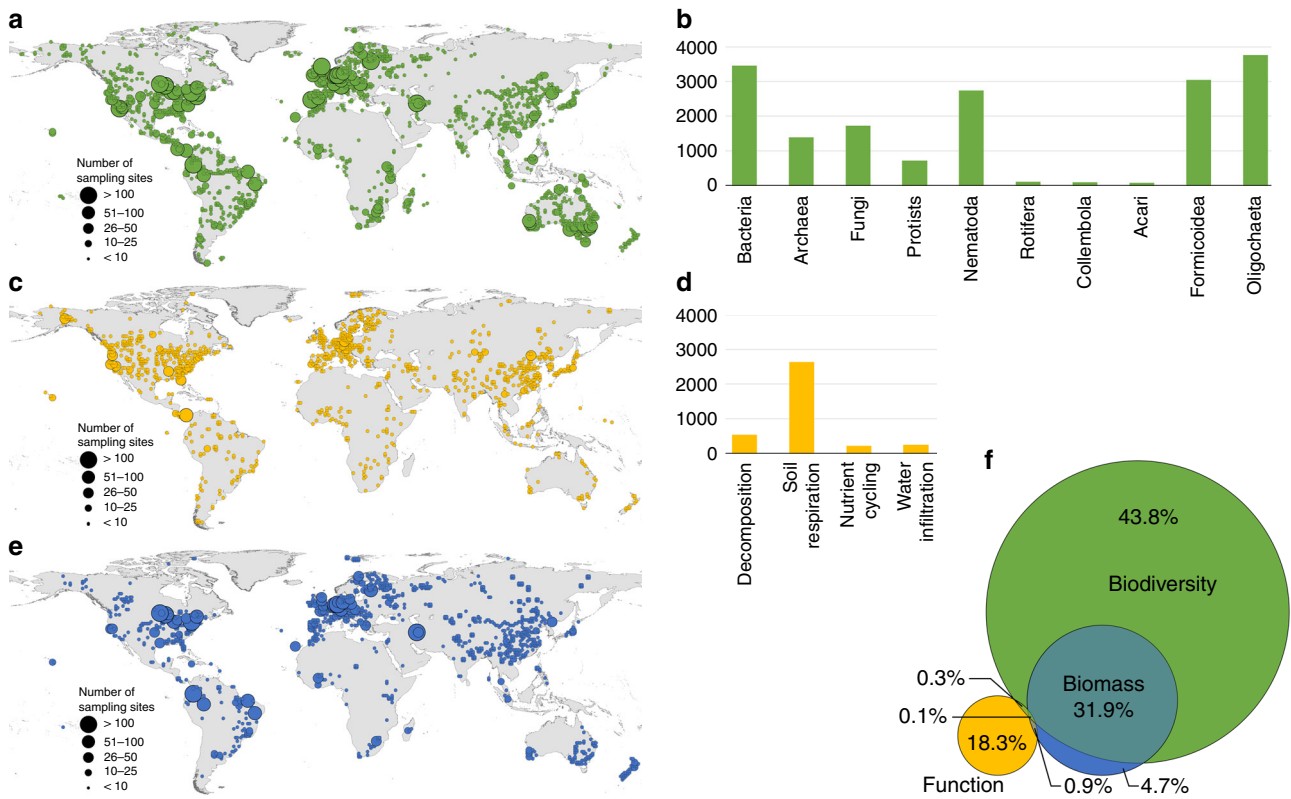

**Fig. 1 Global distribution of sampling sites for soil taxa and soil ecosystem functions. a**, **b** correspond to the global number of individual sampling sites for each soil taxon, **c**, **d** to the distribution of ecosystem functions, and **e** to the distribution of samples with biomass data. The venn diagram (**f**) indicates the proportion of sampling sites for soil taxa (in green), functions (in yellow), and biomass (in blue), and the 0.3% (N = 63) of overlap between biodiversity and function data points (this number does not mean that soil biodiversity and function were assessed in the same soil sample or during the same sampling campaign; i.e., there are thematic or temporal mismatches, see Supplementary Fig. 11 for more details), relative to the total number of sampling sites covered by the studies. The maps show the overall spatial distribution of sampling sites for all taxa (**a**) and soil ecosystem functions (**c**). The size of the circles corresponds to the number of sampling sites within a 1° grid ranging from <10 to >50. All supporting data at: 10.6084/m9.figshare.12581306.

sampling initiatives[46] or a combination of both[47]. Still, most of the analysis made so far lack a balanced representation of the world's ecosystems with most observations concentrated in temperate systems (Supplementary Fig. 4) and suffer from methodological limitations rising from the comparison of multiple methods and sampling schemes.

Soil ecosystems are by nature very heterogeneous at local scales[48]. Having a small and scattered number of sampling sites for both soil functions and taxa (Supplementary Fig. 11) limits the power of current global analyses to evaluate macroecological relationships between soil biodiversity and ecosystem function, particularly for nutrient cycling and secondary productivity, which have strong local inter-dependencies[49]. In fact, from the five functions assessed here, there is a clear concentration of studies on soil respiration, accounting for 78.8% (N = 2616) of all function records (Fig. 1d; see Supplementary Figure 3 for more detail).

We found a lack of matching data for soil biodiversity and multiple ecosystem functions in current global datasets. Due to the dependency of these and other soil functions on biodiversity[2,50], being able to deepen our understanding of the strength and distribution of expected biodiversity and ecosystem function relationships is a needed step to better inform management and policy decisions[51]. In this context, only 0.3% of all sampling sites have an overlap between biodiversity and function datasets (corresponding to 67 sampling sites, Supplementary Fig. 11), with a non-systematic coverage of just a few taxa and functions across sites. Nowadays, macroecological studies on aboveground biodiversity and ecosystem functioning[19,42,52–55] rely on data mobilization mechanisms that allow data to be reused to address multiple research questions. By contrast, apart from some taxonomic groups (i.e., bacteria and fungi) soil macroecological studies based on observational data have a very small degree of overlap and remain conditioned by poor data sharing and mobilization mechanisms[56–58]. Two exceptions to the latter are recent studies on nematodes[44] and earthworms[43] that relied on large synthesis of locally available data, paving the way for more efforts in synthesizing soil diversity data. Nevertheless, these studies have to cope with large arrays of methodological approaches that limit some macroecological analysis (e.g., compositional turnover studies). Also, although the large number of sampling locations, these new datasets further reflect the current biases in macroecological findings by being skewed toward temperate systems (Supplementary Fig. 12).

We also found that most studies are based on single sampling events, i.e., without repeated measurements over multiple years or long time periods for the same sampling sites. Being able to study how communities and functions change over time is essential for assessing trends in key taxa and functions, and their vulnerability to global change[17]. Our global survey suggests that such information is almost nonexistent in large-scale soil biodiversity and ecosystem functions studies. Thus, for most soil communities and functions, although local studies exist[59,60], understanding the global trends and the implications of global change drivers and scenarios is difficult and limited by the absence of globally distributed and temporally explicit observational data.

**Ecological blind spots**. Overall, both soil biodiversity and eco-system function variables reveal a high degree of spatial clustering across global biomes: temperate biomes (especially broadleaved mixed forests and Mediterranean) contain more sampling sites than tundra, flooded grasslands and savannas, mangroves, and most of the tropical biomes, with the exception of moist broadleaf forests (Supplementary Fig. 4). This spatial clustering is even more pronounced in studies of ecosystem functions, with temperate systems being overrepresented with 62% of all sampling sites, while the rest of the globe has scattered information on soil conditions. This likely reflects differences in funding availability and research expertize across countries[27,61]. In fact, for taxa like Collembola and Nematoda (despite a large number of sampling sites), most sampling sites are concentrated in temperate regions, with very few documented in other regions. This imbalance results in quite accurate predictions of soil biodiversity for temperate regions, but with high standard errors elsewhere, which is further enhanced when coupled with non-spatially-stratified algorithms[43,44,62].

Furthermore, the availability of soil biodiversity and function data is especially scarce or even non-existent in tropical and subtropical regions (see Supplementary Fig. 4 for more details), which are among the most megadiverse places on Earth, montane grasslands, and hyper-arid areas. In many cases, local experts and study sites may exist, although their contributions are often not included in macroecological studies and therefore their environmental characteristics are not covered. At the same time, for many of the best-represented regions in the globe, there is rarely a complete coverage of soil taxa and functions, with records often being dominated by one or two densely sampled taxa (e.g., bacteria and fungi) or functions (e.g., soil respiration).

The range of environmental conditions currently described within soil macroecological studies is necessary to understand the relationship between soil biodiversity, ecosystem functions, and key environmental conditions (e.g., the known relationship between bacteria richness and pH[62] or the dependence of soil respiration on temperature[63,64]). In this context, the complete range of soil carbon levels existing on Earth is not well covered, with soils of very high and low carbon contents (Fig. 2a) being underrepresented compared with their global distribution. The same applies to soil type, with only a fraction of soil types being well covered (i.e., acrisols, andosols, cambisols, kastanozems, luvisols, and podzols), while others are underrepresented or completely absent (e.g., durisols, stagnosols, and umbrisols; Fig. 2o). In contrast, our study identified over- and under-represented environmental conditions in soil biodiversity and function studies (Fig. 2). For example, some soil properties are well represented across studies, such as soil texture (i.e., sand, silt, and clay content) and pH, with the exception of extreme ranges (e.g., pH > 7.33 or silt content <19%).

In contrast to soil conditions, climate variability is poorly covered in soil biodiversity and function studies, with several climatic ranges being almost completely missing (Fig. 2f–k). These include low and high potential evaporation/aridity areas and those with high climate seasonality, low precipitation, and extreme temperatures (i.e., very hot and very cold systems), with no overall differences in coverage between biodiversity and ecosystem function studies. Drylands, for example, cover ~45% of the land surface[65] and have been shown to be highly diverse in terms of soil biodiversity and with strong links to specific ecosystem functions[24,66], but are often underrepresented (although some studies specifically target them[10,12]). Climatic conditions (current and future) have strong influences on both soil organisms[60] and functions[63,67,68]. As such, assessing a wide range of these conditions, including climatic extremes, is

fundamental to describe the complex dynamics of soil systems. This issue is further exacerbated when looking at specific climate combinations (Fig. 3c), where 59.6% of the global climate conditions are not covered by any of the studies considered.

Although representing a major driver of soil biodiversity and function[4], land-cover based studies have shown different responses across groups of soil organisms[59,69,70] and specific functions[71,72]. While, in general, land cover types are well covered, sites in the proximity of urban areas are disproportionately overrepresented (Fig. 2n). Climate and soil properties shape soil communities worldwide[62]. Nevertheless, anthropogenic disturbances, particularly those related to land-use change and intensity, have important impacts on these soil communities and their functional performance. Lichens, mosses, and bare areas have been neglected, and shrublands are not well represented in ecosystem function assessments. These gaps may have important implications, particularly when they correlate with understudied ecosystems like drylands or higher latitude systems that may harbor high biodiversity[66], but for which patterns are mostly unknown. In this context, the present analysis indicates that low diversity areas (here represented as plant richness[73]) are absent from most studies or poorly represented, with the focus being mostly on higher diversity areas. Concurrently, it has been suggested that there may be important mismatches between above- and belowground biodiversity across the globe[74], i.e., there are huge areas where aboveground biodiversity does not well predict belowground biodiversity. Although most macroecological studies point in this direction (mismatch between above- and belowground diversity), there are still important dependencies between above- and belowground diversity[75,76]. These dependencies can be functional and, in the case of some groups (e.g., like fungi), can also increase diversity and biomass, with positive effects on soil carbon storage[77].

When looking at how belowground studies cover global environmental conditions (Fig. 3a, c), important spatial gaps are observed. Although most soil-related variables are well covered across studies, the same does not apply when looking at aboveground diversity (see Supplementary Fig. 10), which shows a very good coverage in forest and crop areas with above-average plant richness in mid to low elevations, while other environmental combinations are underrepresented. Overall, while it is unreasonable to expect all macroecological studies to cover all possible soil conditions our results show that most studies have, on average, a coverage below 50% across global regions, with the exception of Central and west Europe and Caribbean (for both biodiversity and function) and Central and North-East Asia and North and South America (for ecosystem functions). All other regions show a systematic poor coverage across soil macro-ecological studies with North Africa and West Asia having the lowest average environmental coverage (Fig. 3b, d).

Although temperate regions (e.g., Central and West Europe) have the highest average environmental coverage across soil macroecological studies, within these regions many environmental combinations are not properly covered, particularly areas of high altitude with low pH and high carbon content (see Fig. 3AF, BF). Worryingly, regions or countries considered to be mega-diverse (i.e., at least from an aboveground diversity point of view) are systematically poorly represented across soil macroecological studies (Fig. 3aB, aE, aD, cB, cE, cD).

Many of the reasons and drivers of existing data gaps have already been discussed in recent literature for aboveground systems[16] (e.g., accessibility, proximity to large cities, etc.). In the case of soil biodiversity and ecosystem functions, these blind spots are further reinforced because of the lack of standardized protocols for acquiring biodiversity and ecosystem function data. This translates into an absence of comparable data, which is even

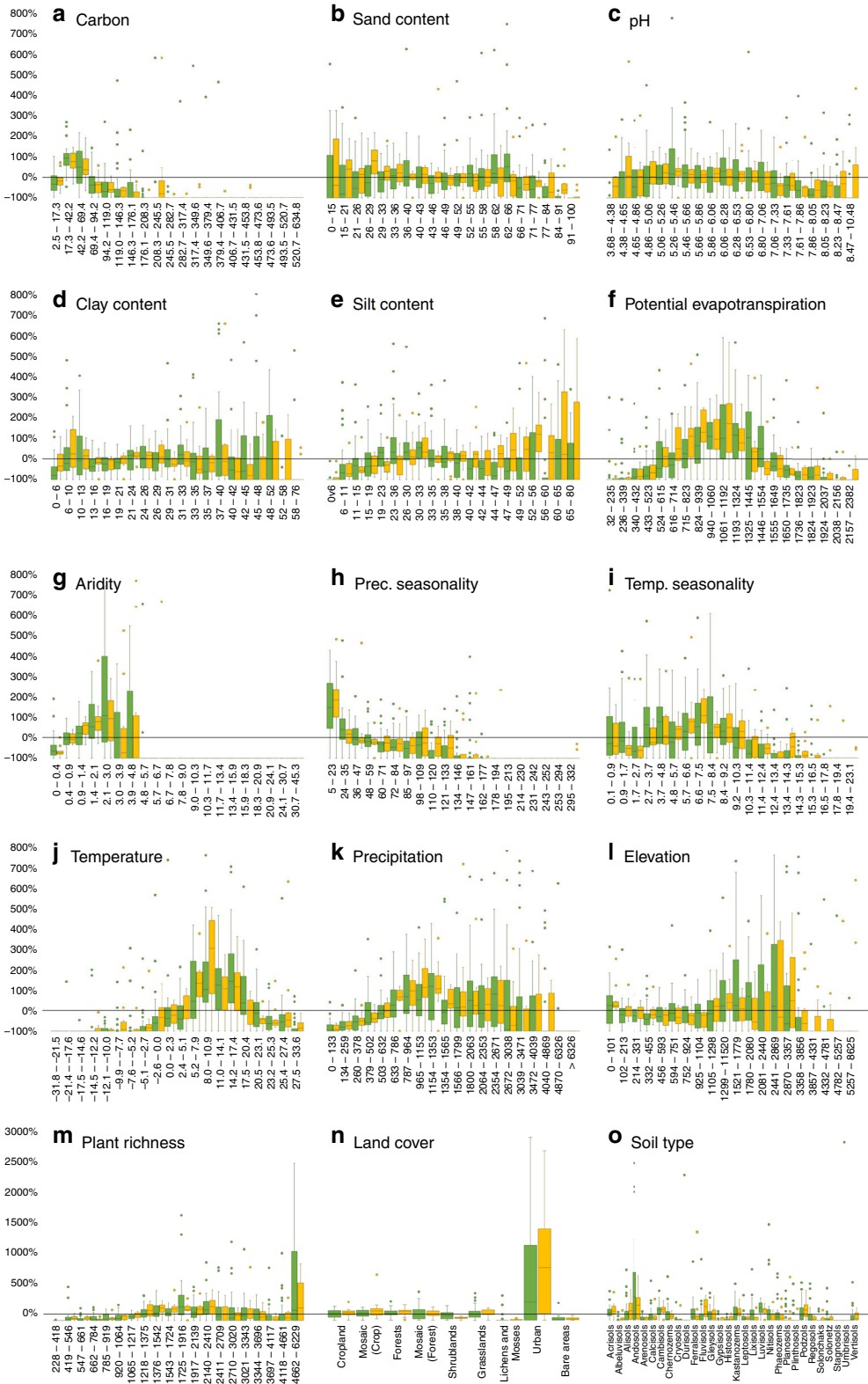

more pronounced than in other systems[16,78]. Nevertheless, there is a continuous movement towards improving data mobilization and international collaborations that could help overcome these issues if steered in the direction of underestimated taxa and/or functions identified here[79].

In a changing world where soil biodiversity shifts are being systematically reported[80–82], and where current forecasts are pointing to increases in land-use intensity[83,84], desertification[85], and rapid climate change[86–89], understanding if and to what extent biodiversity changes are happening in soil communities is of high importance. This is particularly relevant to assess causal effects between changes in biodiversity and ecosystem function (e.g., are changes in biodiversity occurring because of changes in function, paired with them, or despite them, and vice versa), which is even more relevant if key ecosystem functions (e.g., carbon sequestration) are the subject of evaluation.

**Fig. 2 Global soil ecological blind spots.** Values (y-axis) correspond to the percentage of sites per study when compared with the global percentage distribution (e.g., a value of 20% means that a given study overrepresents a given environmental variable by 20%, when compared to the global distribution of that same variable). Soil biodiversity studies in green ($N = 35$) and ecosystem function studies in orange ($N = 12$). **a** soil carbon (g soil kg$^{-1}$)[127]; **b** sand content (%)[127]; **c** soil pH[127]; **d** clay content (%)[127]; **e** silt content (%)[127]; **f** potential evapotranspiration (mm/day)[128]; **g** aridity index[128]; (**h**) precipitation seasonality[129]; (**i**) temperature seasonality[129]; **j** mean annual temperature (°C)[129]; **k** mean total precipitation (mm)[129]; **l** elevation (meters)[130]; **m** vascular plant richness[73]; **n** land cover[131]; and **o** soil type[127]. The zero black line corresponds to a situation where the proportion of sites in a given class within a study matches the global proportional representation of the same class. Although outliers were not eliminated, for representation purposes these were omitted >800% between panels **a**–**l** and >3000% for panels **m**–**o**. The class intervals of each continuous variable were obtained based on a natural breaks (Jenks) classification (20 classes). Each barplot (quantile distribution) represents the proportional number of sampling sites covering a particular class when compared to the global distribution. In panel (**n**) mosaic (crops) represent small scale landscapes dominated by crops, while mosaic (forests) represent small scale landscapes dominated by forests.

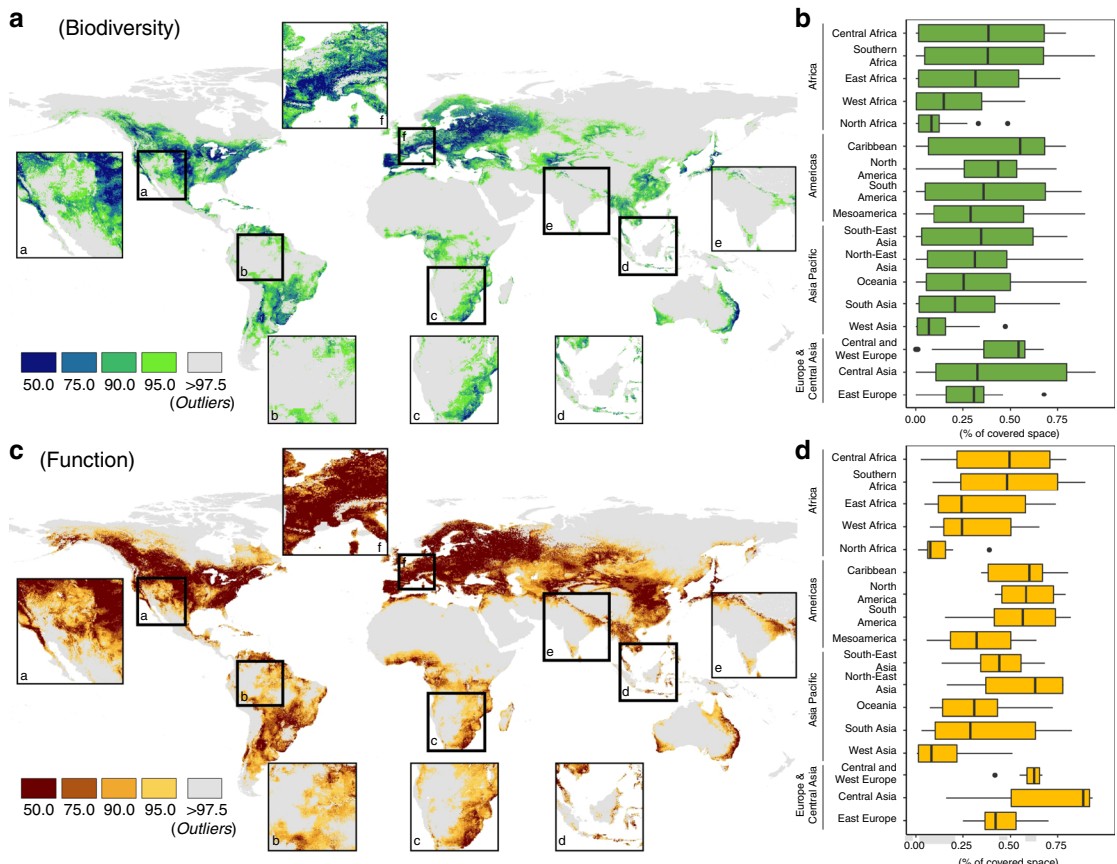

**Fig. 3 The extent to which main soil environmental characteristics are covered across macroecological studies.** Colors (in **a** and **c**) correspond to the average $\chi^2$ values across all studies considered and environmental conditions calculated based on Mahalanobis distance[123–125] (gray color corresponds to outlier conditions: see Methods for more details) within: **a** and **b** corresponding to the biodiversity studies and **c** and **d** to ecosystem function studies. **a** and **c** correspond to the spatial distribution of the $\chi^2$ values to 0.50, 0.75, 0.90, 0.95, and 0.975 break points. **b** and **c** correspond to boxplots (quantile distribution) of the percentage of area covered (<0.975 $\chi^2$) by each study considered across the different IPBES regions. Results show that most studies have, on average, a coverage below 50% of all the regions in the world, with the exception of Central and west Europe (**f**) and Caribbean (for both biodiversity and function), Central and North-East Asia, and North and South America (for ecosystem functions). **a**–**f** correspond to zooms on specific areas of the globe. All supporting data at: 10.6084/m9.figshare.12581306.

**Challenges to move beyond blind spots.** Filling the knowledge gap on large-scale temporal trends in soil biodiversity and ecosystem function cannot be achieved without spatially explicit studies based on resampled locations. This could be done with a standardized global monitoring framework that is recognized and supported by a large number of countries, which currently does not exist. Given the strength of recognized soil taxa interactions[90], biodiversity and ecosystem function relationships[24], and above-belowground interactions[91], these large-scale monitoring activities and research studies should consider going beyond traditional single taxa/function approaches and collect information on the

multiple dimensions of soil ecosystems[28], while at the same time expanding/supporting surveys to cover the blind spots of soil macroecological research (Fig. 3).

Across all soil taxa and functions, the geographical and ecological blind spots identified here often emerge from a number of obstacles specific to soil ecology[79] (see summary in Table 1). Soil macroecologists face many challenges and constraints spanning from a lack of methodological standards and scientific expertize in different taxonomic groups[92–94], to limitations caused by the current implementation of the Convention for Biological Diversity (CBD) and the Nagoya Protocol[95,96]. While

**Table 1 Summary of the main obstacles soil ecologists face to create a global soil biodiversity monitoring network and the priority actions to overcome them.**

| Challenges | Researchers Priority actions | Institutions | Policymakers |
|---|---|---|---|
| Legal issues regarding the transport and sharing of soil samples and biological data | Raise the awareness of institutions and decision-makers about the importance of these legal bottlenecks for the development of international research programs. | Develop a legal understanding of the implications of material transfer mechanisms for soil samples and provide support to researchers also by promoting knowledge and expertize exchange. Support and facilitate the establishment of international consortia and bilateral institutional agreements particularly with developing countries | Establish global multilateral solutions and International Treaties focused on soil biodiversity and ecosystem function research. Establish knowledge transfer mechanisms for soil-related research together with the classification of soil samples for research purposes. |
| Scattered literature and lack of mobilization/ systematization of local studies | Invest in data harmonization, synthesis, meta-analysis approaches, data collation, and standardizedr metadata to improve currently available datasets (e.g., through GBIF for soil biodiversity). Publishing under free "Open Access" (OA) licence and/or using preprint platforms or fully OA journals. Define and publish data standards that allow for better data transfer focussing on the methods, reporting in standard units, and best practices for data availability. Increase the focus on understudied soil groups (e.g., collembola, acari, protists, mammals) and functions (e.g., soil aggregate stability, bioturbation, nutrient cycling). Establish effective coordination of current networks to support the development of integrated ecological assessments of the soil realm | Adoption of available data and methods standards[101, 132–136] and support the establishment and maintenance of data repositories and open access policies. | Support open access partnerships (e.g., the German DEAL[137]) to facilitate knowledge transfer and collaboration across countries and researchers from different backgrounds and expertize. Improve the digitally available data on soil biodiversity and ecosystem function by supporting the expansion of current global databases (e.g., GBIF) or the creation of interoperable data infrastructures on soil function data. |
| Lack of temporally explicit information on soil biodiversity and functions | Identify relevant sites - e.g., sites covering a wide range of taxa or functions and/or a high degree of standardization - for resampling. Revisit already sampled sites to obtain temporal measurements of soil biodiversity and ecosystem function. | Institutional support of long-term databases and collections of soils, soil functional data, and soil biological material. | Create funding schemes for strategic long-term research projects on soil monitoring and research (e.g., using the LTER framework as an example[138]). |
| Lack of globally distributed expertize, research funding and infrastructures | Promote knowledge transfer mechanisms and capacity building, especially with developed countries that might see little advantage of being involved in a global network that only offer co-authorship as the main benefit. Setup international workshops, summer schools, or classes with a focus on educating the next generation of scientists on different aspects of soil ecology. | Build on or expand current networks to include knowledge transfer activities, namely on education, methods calibration, sharing research facilities, and taxonomic expertize. | Promote funding flexibility to train and empower researchers across countries and/or regions, also allowing local scientists, particularly in the developing world, to conduct soil biodiversity and ecosystem function research. Establish soil health as a research priority beyond farming areas and with a special focus on ecological conservation of soil organisms and ecosystem functions. |

the first has more immediate, albeit non-trivial solutions (e.g., by expanding the language pool of the researchers and studies included[16,97] and by applying common standards for sampling, extraction, and molecular protocols[98–101]), the latter contains systemic issues that go beyond soil ecology alone. In this context, although the CBD and the Nagoya Protocol were created to protect countries while making the transfer of biological material more agile, numerous states have either not yet implemented effective national "Access and Benefit Sharing" (ABS) laws or have implemented very strict regulations[102,103]. Yet, even after 25 years of the CBD and the ABS framework being in place, the major motivation for a strict national regulation - the anticipated commercial benefits and high royalties from the "green gold" - has not yet materialized[95,104].

Researchers have yet to coordinate a global effort to characterize the multiple aspects of soil biodiversity and function in a comprehensive manner, with the current literature dominated by scattered, mostly local studies focused on specific soil organisms and/or functions. Although here we do not comprehensively assess the potential of local studies to overcome the current blind spots,

other studies[34,35,67] have shown that, with an effort in standardization and data mobilization, local and regional studies add fundamental knowledge and empower local researchers to participate in global initiatives. In fact, several studies not included in this assessment can provide a finer-scale resolution in many areas of the globe[69,105,106]. Nevertheless, their spatial extent systematically coincides with overrepresented areas (e.g., temperate areas), and their taxonomic and functional focus is mostly on the already prevailing taxa (i.e., bacteria and fungi) and functions (i.e., soil respiration), potentially increasing existing biases. This increases the relevance of facilitating data mobilization from regions and, more importantly, environmental conditions that are systematically not covered by macroecological studies. Here a word of caution is needed as scientists from around the world need to publish their findings to progress their careers and/or to obtain research funding. Managing large data sets and having access to global databases on climatic and soil information, is often a privilege for well-resourced research teams (in terms of bibliographic subscriptions, computing power and technical knowledge, and software[107,108]). Given the diversity of global conditions (both scientific and environmental), data mobilization alone is not the solution and needs to be paired with the priority to have more national (and local) surveys across a large number of sites and with a deeper taxonomic level.

In parallel, and given the nature of global change drivers, understanding their influence on local soil communities and ecosystem functioning requires global macroecological approaches that can provide context, predictions, and concrete suggestions to policymakers across the globe. Yet these macroecological approaches will be less effective in providing relevant outputs at national scales if based on data extrapolated from other countries; they would be strongly improved if local data would exist coming from national and local surveys and were made openly available[25,109]. Without more comprehensive studies seeking answers to large-scale soil ecological questions - often involving dealing with multiple scales (temporal and spatial) and a number of thematic and taxonomic depths[74] - it is difficult to deepen soil macroecological knowledge[110]. This is particularly relevant in testing biodiversity and ecosystem function relationships at the global scale, or trying to address specific societal issues (e.g., the attribution of climate and land-use change as drivers of soil ecological change or general biodiversity trends)[17].

Another major challenge is associated with the fact that currently ABS agreements are bilateral. This hinders global soil ecology initiatives, as it requires that providers and receivers need limited (in time and topic) individual material transfer agreements. Thus, for a global initiative, this can amount to hundreds of material transfer agreements. However, there is an increasing quest for global solutions and multilateral systems, such as the International Treaty on Plant Genetic Resources for Food and Agriculture (IT PGRFA; www.fao.org/3/a-i0510e.pdf), or other harmonized best practices examples like the Global Genome Biodiversity Network[96]. As the commercial value of soil organisms is regarded to be zero in situ[111], and as these are mostly ubiquitously distributed at the highest taxonomic level (e.g., bacteria and fungi), soil per se has no commercial value as it does not match the criteria that "provider countries host unique and unmatched biodiversity" of the Nagoya Protocol[112]. Therefore, a global multilateral solution, similar to the examples listed above (e.g., the IT PGRFA), but focused on facilitating the exchange of soil samples to drive basic research on soil biodiversity, taxonomy, and ecosystem functioning, while still safeguarding against the spread of foreign genotypes, is pressingly needed. At the same time, as long as bilateral ABS agreements are required, researchers should engage with local policymakers to enable unrestricted soil biodiversity and ecosystem function research, as was the case for Brazil from 2006 to 2016[113].

**Looking for solutions to unearth global observations**. Globally, soil habitats are under constant pressure from major threats, such as climate change, land use change and intensification, desertification, and increased levels of pollution. Here, we argue for global monitoring initiatives that systematically samples soil biodiversity and ecosystem functions across space and time. Such global initiatives are urgently needed to fully understand the consequences of ongoing global environmental change on the multiple ecosystem processes and services supported by soil organisms (Table 1). This requires that current and future funding mechanisms include higher flexibility for the involvement of local partners from different countries in global research projects. Given that soil ecological research requires cross-border initiatives[79,108] and often expensive laboratory infrastructure, there is a need for flexible funding with proper knowledge transfer mechanisms to sustain global soil macroecological research. Such knowledge will in turn contribute to advancing our understanding of macroecological patterns of soil biodiversity and ecosystem function, thereby fulfilling national and global conservation goals[111,114,115].

Considering the current pool of literature, improving the digitally available data on soil biodiversity and ecosystem function should be a top priority that can be supported by systematically mobilizing the underlying data[116] in already existing open access platforms (e.g., GBIF). Achieving this goal on shared knowledge and open access data will return benefits beyond making global soil biodiversity surveys possible. It will allow local researchers to expand their own initiatives, create a more connected global community of soil ecologists, bypassing publication and language limitations, and potentially open doors in countries that may otherwise be reluctant in sharing their soil biodiversity data[27].

In parallel, coordinated sampling strategies based on standardized data collection and analysis are needed to improve soil macroecological assessments. From our results, it is clear that most, if not all, studies look at only a fraction of the soil realm without much spatial and thematic complementarity of global environmental conditions. Also, the small overlap between biodiversity and functional studies indicates that most community assessments disregard the ecosystem functions that these provide and vice versa, prompting a call for more complex approaches that can show potential links and global ecosystem services. Our study helps to identify global target locations and biomes, which need to be given priority in future surveys. Future sampling strategies would greatly benefit from coordinated sampling campaigns with biodiversity and function assessments at the same locations and ideally from the same soil samples to improve the current spatial-temporal resolution of data on soil biodiversity and ecosystem functions.

These two complementary pathways (i.e., data mobilization and sharing of current literature and a globally standardized sampling) if done in a spatially explicit context, and following standardized protocols, could ultimately inform predictive modeling frameworks for soil ecosystems to track the fulfillment of global/national biodiversity targets, policy support, and decision-making. Taken together, our study shows important spatial and environmental gaps across different taxa and functions that future macroecological research should target, and a need to collect temporal datasets to explore if current aboveground biodiversity declines are found in belowground taxa. With the identification of global spatial, taxonomic, and functional blind spots, and the definition of priority actions for global soil macroecological research[74], our synthesis highlights the need for action to facilitate a global soil monitoring system that overcomes the current limitations.

## Methods

**Literature selection and data processing**. We created a dataset by collecting published literature on macroecological studies of soil biodiversity and ecosystem

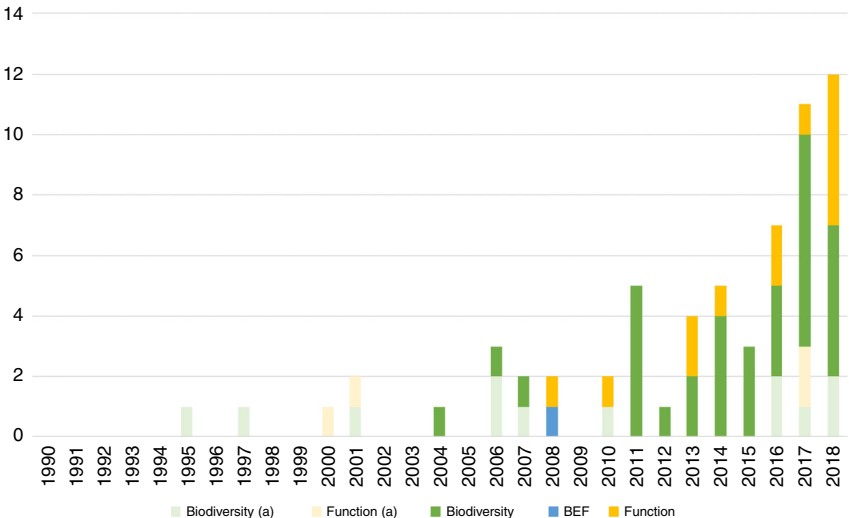

**Fig. 4 Accumulated number of papers screened for this analysis.** Studies were classified in soil biodiversity, function or biodiversity, and ecosystem function (BEF), according to the subject of the study (see Table 2). **a** corresponds to the number of studies that were not included due to the underlying data not being suited for this analysis (e.g., based on national level information) or data availability issues. Overall, ~72.6% of the total number of studies identified as suitable were included in the analysis ranging from 2004 to 2018.

functions. For this literature search, we conducted a search in the Web of Knowledge database in November 2018 within papers published between 1945 and 2018 using the keywords: (Global* OR Continental OR latitud*) AND (soil* OR belowground) AND (*function* OR *diversity OR organism* OR biota OR animal* OR invert* OR fauna*) AND distribution AND (*mycorrhiz* OR microb* OR nematod* OR bacteria* OR ant* OR fung* OR invertebrate* OR earthworm* OR protist* OR eukaryot* OR collembola* OR rotifer* OR archaea OR formic* OR mite* OR termite* OR arthropod* OR respiration OR decomposition OR nitrogen-cycling OR nutrient cycling OR water infiltration OR aggregate* OR bioturbation OR biomass). These keywords were selected to encompass the maximum number of published studies, which often use a variety of expressions for describing soil biodiversity and function. Additionally, we included studies found in the references of the papers returned by the database, as well as opportunistically gathering additional studies, such as from personal bibliographic databases of global soil studies.

The initial Web of Science search returned 1203 studies, which were screened for the following three inclusion criteria: (1) studies dealing with soil taxa and/or soil ecosystem functions; (2) studies spanning more than one continent with more than single locations in each continent (the Pacific islands not included in any particular continent were counted as an individual continent, contributing to the inclusion of some studies); and (3) studies that span across an entire continent (i.e., a study focussing only on Europe would not be included but if larger scales would be assessed - e.g., across Eurasia - the study would be included). From these, the number of studies was reduced to 58 that, after being complemented with additional studies, resulted in a dataset with 62 studies dealing with processes and patterns of soil biodiversity and/or soil ecosystem function across global gradients ranging from 1995[117] to 2018[118] (Fig. 4, Table 2).

The 62 studies selected were then classified according to the taxon and/or function that was subject of the paper and screened for the availability of point coordinates for the sampling sites underlying the study (Table 2). We obtained the locations of the sampling sites that supported each manuscript, either by using published data or by contacting the corresponding authors of the publications. Papers including only national or regional level information were not included in this analysis. This was not done because of any consideration about the quality of the study nor related to bad reporting, but rather because of the lack of comparable data across studies. Overall, ~72.6% (45 studies) of the total number of studies identified were included in the analysis (see Table 2 for more details). This allowed us to cover ten different taxa and five (from a total of seven) ecosystem functions with important relevance for biogeochemical cycles[2,119,120] and the supply of key ecosystem services[1,3] (Fig. 1; Table 2).

Available pairs of coordinates were then georeferenced and projected to WGS84 to create a dataset comprising all individual studies. The geographic coordinates of the sites were included based on the data provided by the studies and no spatial corrections were made (Fig. 1). In the manuscript, sampling sites refer to unique X, Y pairs of coordinates that represent individual locations. As a final step, sampling sites located outside of the terrestrial scope of this paper - which excludes Antarctica and Greenland - were removed from the subsequent analysis.

**Completeness and estimation of global representation.** One of the main objectives of this paper is to describe the ability of current soil macroecological

| Table 2 List of studies included in the current assessment. | | |
|---|---|---|
| **Soil biodiversity**[a] | **Included studies** | **Not included studies** |
| Bacteria | 10, 14, 35, 66, 70, 120, 139–145 | 119, 146–149 |
| Archaea | 70, 120, 141, 142, 150 | 146, 149 |
| Fungi | 45, 110, 141, 142, 145, 151–156 | 119, 146, 157–159 |
| Protista | 70, 160, 161 | 117, 162–164 |
| Nematoda | 13, 44, 165, 166 | 119 |
| Rotifera | 166, 167 | – |
| Collembola | 168 | 119 |
| Acari | 166, 168 | 119, 169 |
| Formicoidea | 34, 46, 170 | – |
| Oligochaeta | 43, 46, 118, 166, 168, 171 | – |
| *Soil functions*[a] | – | – |
| Decomposition | 67, 72, 168 | 172 |
| Soil respiration | 63, 71, 116, 173–175 | 176 |
| Nutrient cycling | 177 | – |
| Water infiltration | 178, 179 | – |
| Bioturbation | – | 180 |
| Soil aggregate stability | – | 181 |

[a]Given the thematic scope of some studies, an individual study can be included in more than one taxon/function.

research to capture the diversity of conditions affecting the soil realm, here described as the bulk of characteristics encompassing the soil (including physical and chemical properties), climate (including properties that affect soil conditions e.g., related to soil humidity and temperature), geomorphology (including global topographic properties), and aboveground diversity[1,41]. With this definition, we identified 15 environmental and diversity descriptor variables of the soil realm (Table 3) that we used to characterize how the overall sampling locations for each study capture the global environmental and aboveground diversity scope. Initial calculations were made using ArcGIS and ArcPy at the original resolution of the different datasets although, for the final spatial mapping and integration, all datasets were harmonized to an ~1 km² (at the equator) resolution using a resampling algorithm without changing the original values - i.e., focussing only on pixel disaggregation with a nearest neighbor classifier.

For each study, we examined how these sample-based distributions capture the diversity of global conditions with the purpose of finding the "blind spots" of global soil ecosystem research corresponding to underrepresented areas of low environmental and aboveground diversity representation. To do this, we compared the global histogram of each of these environmental and aboveground diversity variables (Table 3) with the one obtained using the sampling sites for every

**Table 3 Environmental variables defining the soil realm.**

| Environmental variable | Dataset | Reference |
|---|---|---|
| Soil carbon | SoilGRIDS - global soil information based on automated mapping | 127 |
| Soil pH | SoilGRIDS - global soil information based on automated mapping | 127 |
| Clay content | SoilGRIDS - global soil information based on automated mapping | 127 |
| Sand content | SoilGRIDS - global soil information based on automated mapping | 127 |
| Silt content | SoilGRIDS - global soil information based on automated mapping | 127 |
| Soil type | SoilGRIDS - global soil information based on automated mapping | 127 |
| Mean annual temperature | CHELSA - Climatologies at high resolution for the earth's land surface areas | 129 |
| Mean annual precipitation | CHELSA - Climatologies at high resolution for the earth's land surface areas | 129 |
| Temperature seasonality | CHELSA - Climatologies at high resolution for the earth's land surface areas | 129 |
| Precipitation seasonality | CHELSA - Climatologies at high resolution for the earth's land surface areas | 129 |
| Aridity | CGIAR-CSI - Global aridity database | 182, 183 |
| Potential evapotranspiration | CGIAR-CSI - Global potential evapotranspiration database | 182, 183 |
| Elevation | GMTED2010 - Global Multi-resolution Terrain Elevation Data | 130 |
| Land cover type | ESA CCI - Global Land Cover database | 131 |
| Plant diversity | Global plant diversity | 73 |

particular study. These histograms were classified using a natural breaks (Jenks) method[121], with the exception of land cover and soil type for which the original categorical classification was maintained. This procedure allowed us to identify particular ranges of environmental and diversity conditions that are overrepresented in current literature and not assume total coverage by just looking at the spatial distribution of sampling sites. For example, having one sampling site in the tropics may wrongly give the impression that tropical regions are covered when in fact this sampling site can only cover a very small range of the total tropical spectrum. Given the small scale differences in soil communities, by contrast with other aboveground taxa, this overrepresentation of specific conditions in detriment of others is of the utmost importance as it can produce important interpretation biases and knowledge limitations.

Second, for each study, we overlaid the spatial representation of all variables to include the mean and median distribution and the standard deviation across environmental variables. In order to have taxonomic and functional representations (e.g., for Bacteria, fungi, decomposition, etc.), we replicated the procedure by including all the studies identified for each taxon and function. In parallel, we used global biomes[122] as a spatial stratifier to understand global biases in soil biodiversity and function data and representation. Finally, to assess the representation of soil diversity and climate conditions, it is not enough to evaluate them independently of each other. Therefore, we combined the previously classified variables (the spatial distributions of each classified variable can be found in Supplementary Figs. 5–9) in three different groups (Supplementary Fig. 10): (a) land cover (including the combination of land cover, plant diversity and elevation); (b) soils (including the combination of organic carbon, sand content, and pH); and (c) climate (including the combination of mean precipitation and temperature, and their seasonality). We finally overlayed these mapped combinations with the distribution data from each study in order to understand which combinations have the highest number of redundant studies, i.e., more than one study covering the same environmental combination.

Finally, we applied a statistical method based on the calculation of a multidimensional distance (considering all continuous environmental variables) based on mahalanobis distance[123,124]. For each study, we calculated and mapped the Mahalanobis distance of all locations to the center of the observed distribution given by the sample distribution of each study. This distance is useful to detect outliers in point cloud distributions that are assumed to follow a multivariate Normal distribution[123,124]. When each of the variables is normally distributed, the Mahalanobis distance follows a $\chi^2$ distribution with d degrees of freedom, where d corresponds to the dimension of the multidimensional space (i.e. the number of environmental variables used)[123,124]. Figure 3 uses a color gradient to indicate the quantile of the Chi squared distribution with 13 degrees of freedom that each xy coordinate belongs to Mallavan et al.[125].

We acknowledge that sampling schemes should in the future consider the amount of diversity represented in each region and, therefore, match the sampling effort with estimates of completeness coverage of actual composition values[126]. Nevertheless, due to the lack of access to the actual diversity and functional data (apart from publicly available datasets) and, more importantly, to the assumption that current global soil biodiversity hotspots are either poorly characterized (due to data constraints) or fail to account for the diversity of environmental conditions present in the globe, we implemented the current approach as an alternative to highlight the environmental coverage of current macroecological studies. In the future, with more studies shedding more light on the global distribution of soil biodiversity, studies and monitoring initiatives should probably focus on approaches that use the "novelty" of each new sample to infer how (in)complete the description of biodiversity is[126].

As the current analysis was conducted as a global analysis without a regional segmentation (e.g., without partitioning the analysis per continent), sampling sites in one continent (e.g., sampling sites covering high altitude areas in Europe) may influence the coverage in other continents (e.g., may identify areas in North and South America as being partially covered). Although this may lead to the overrepresentation of some areas (and consequent underrepresentation of others), our main intent is to understand how current macroecological literature is capturing the diversity of global soil conditions.

**Reporting summary**. Further information on research design is available in the Nature Research Reporting Summary linked to this article.

## Data availability
All data is available as a single dataset including all soil biodiversity and function locations with reference to the respective manuscripts from which they were extracted. It can be found here 10.6084/m9.figshare.12581306 and here 10.6084/m9. figshare.12581306.

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

## Acknowledgements

This manuscript developed from discussions within the German Centre of Integrative Biodiversity Research funded by the German Research Foundation (DFG FZT118). CAG and NE acknowledge funding by iDiv (DFG FZT118) Flexpool proposal 34600850. C.A.G., A.H.B., J.S., A.C., N.G.R., S.C., L.B., M.C.R., F.B., J.O., G.P., H.R.P.P., M.W., T.W., K.K., and N.E. acknowledge funding by iDiv (DFG FZT118) Flexpool proposal 34600844. N.E. acknowledges funding by the DFG (FOR 1451) and the European Research Council (ERC) under the European Union's Horizon 2020 research and innovation programme (grant agreement no. 677232). Finally we would like to acknowledge the contribution of all the authors that provided their datasets for analysis within this paper. Open access funding provided by Projekt DEAL.

## Author contributions

C.A.G., A.H.B., J.S., A.C., N.G.R., S.C., L.B., M.C.R., F.B., J.O., G.P., H.R.P.P., M.W., T.W., K.K., F.T.M., B.S., and N.E. conceived and designed the study; C.A.G., A.H.B., J.S., A.C., N.G.R., S.C., L.B., M.C.R., F.B., J.O., G.P., H.R.P.P., M.W., T.W., K.K., F.T.M., E.C., C.M., A.O., D.H.W., M.D.B., R.B., D.C., T.G., B.S., and N.E. contributed to the data acquisition and all authors had substantial contributions to the interpretation of the data, writing, and review of the final manuscript.

## Competing interests

The authors declare no competing interests.
