## [Peer Review File · Nature Communications]

Reviewers' Comments:

Reviewer #1:

Remarks to the Author:

This study provides an overview on potential knowledge gaps in soil biodiversity and ecosystem functioning. As the authors acknowledge, the majority of the studies available in the literature merely focus on microbes, and there is an overall lack of research studies dealing with the temporal variability of many other soil organisms across many soil types, land covers, and biomes. However, the manuscript does not provide enough novel information to make a significant progress that could increase this understanding. Overall, I found the study to be very descriptive and lacking more in-depth analyses of the data extracted from the literature. For example, I would have expected to see (from what we have) whether it is possible to find any meaningful patterns in soil biodiversity and ecosystem function at global scales. This would have allowed a more critical and detailed outcome.

General comments:

The fact that studies dealing with soil biodiversity and ecosystem functioning at large scales are a relative recent research field, the selection of the studies appears to be strongly biased and not properly justified. For example, due the scanty number of macroecological studies it would have been useful to also include those individual studies focusing at national levels. In addition, using 1-3 studies to make conclusions about metazoans and four soil functions (according to Table 2 in the methods) is not appropriate. In fact, the maps provided in Figs. S4 are misleading because if I am right, the authors only included one study to draw the maps of Collembolans (Fig. S2a/g), nutrient cycling (Fig. S2b/c) and water infiltration (Fig. S2b/d). In addition, also according to Table 2 in the methods section, the same study (e.g., 151 and 153) have been used to draw different maps.

In the abstract, the authors claim that > 11,000 sampling sites were included, but the actual analyses (and therefore the conclusions) are only based on 45 studies. According to the text, the initial number of sites (11,065) consisted of 7,631 sites reporting soil biodiversity data and 3,497 sites with information on soil functions (L131-133). Did all these sampled sites were obtained from just 45 studies (L757-758)? In L168 it is stated in that 63 sites contained both types of information, should not they be presented to the reader in a map? This would have given a more clear idea of where this type of information is concentrated.

And, did all the soil biodiversity sites have information on several soil organisms or just specific groups?

I am not very convinced that data are mostly absent from most tropical areas (L102) when one of the first international initiatives Tropical Soil Biology and Fertility Programme in the 80s was focused on that particular biome. I was also surprised to see that The International Biological Program (IBP), an effort between 1964 and 1974 to coordinate large-scale ecological and environmental studies, was not even mentioned. It is also surprising that one charismatic group of tropical systems, termites, were not included or mentioned in the paper. And what about all the soil atlases produced by FAO, USDA and ESDAC that provide abundant soil data?

In addition to these large scales projects, there are currently several databases that continuously archive soil organisms and/or soil data, such as Edaphobase, DriloBase, Collembola database, etc. Therefore, I would have expected that the first effort should be the incorporation of all this information in a study that intends to identify knowledge gaps. In addition, before we care about macroecological gaps, should we not aim for a better knowledge of soil biodiversity when it has been stated that we only know a small fraction (e.g. in the macroecological study by Decaens 2010)?

More specific comments:

In several parts of the text there is an overemphasis of the described findings; for example, the use of the term "significant" is misleading when no statistical tests have been applied to the data reported.

Some kind of description/justification is needed to explain how the class intervals for each of the variables depicted in Fig. 2 were established. Were they based on a literature survey or on a specific criterion? Also in this multi-panel figure, Fig. 2n clearly shows that urban areas are overrepresented, what does this mean in terms of number of studies analysed? How can you be sure that climatic, soil and above-ground variables are the main determinants of edaphic life and not anthropogenic disturbances?

The same can be said for the other charts, how many studies reported these values? In several of these charts, 20 intervals were established, meaning that 2 studies were represented in each one... Also units for Figs. 2f-n? Vascular plant richness is also included to describe "global soil ecological blind spots"

(Fig. 2m), but in several parts of the text it is stated that no links between above- and below-ground biodiversity has been found, so which are the reasons behind using this metric as potential influential factor explaining knowledge gaps in soil biodiversity and ecosystem functioning?

I do agree that lack of standardized protocols hinders our understanding, so how was this solved in this study when compiling the data from the different studies? For example, measuring C content by loss of ignition, dichromate-oxidation method, etc. could render different results, as well as different methods to extract soil biota. And what about the use of different soil classifications when plotting soil types?

A justification is also needed to explain the reasons behind combining just three variables when trying to assess which environmental characteristics are used in macroecological studies (Fig. 3). Since that according to Fig. 2 and the explanations given in the text, soil carbon, land cover are biased, should it not be better to illustrate the combination of variables for which we have good coverage? Why was not soil type included? This information is usually available in many published soil atlases...

I like the overview of the main obstacles that soil ecologists face when trying to produce/complete soil surveys and that has been summarised in Table 1. In my opinion, scientific challenges are also important. Despite the authors trying to convince the reader that data sharing and making data available to be re-used again could be the solution to decrease the number of knowledge gaps, I feel that this is only one side of the story. Scientists from around the world need to publish their findings to progress in their careers and/or to obtain research funding. Managing large data sets, having access to global databases on climatic and soil information is often a privilege for rich research teams (in terms of bibliographic subscriptions, computing power and technical knowledge, and graphical software). The text in L365-371 gives the impression that "data mobilisation" should be a priority and that by embracing open access and connecting the global soil community, we could fulfil all those missing knowledge gaps. However, due the current taxonomic deficit for soil biota, the priority should be to have more national (and local) surveys to increase the soil biodiversity information across a large number of sites and to a deeper taxonomical level. Global research projects (L359) would be ideal but unrealistic in many practical terms.

Should not the authors acknowledge the authors who helped when obtained the

locations of the sampled sites?

This is an interesting study that could attract many potential readers if it could provide a clearer message and quantitative analyses (including statistical tests and not just maps) of the available data.

Reviewer #2:

Remarks to the Author:

The manuscript from Guerra et al. "Blind spots in global soil biodiversity and ecosystem function research" represents to me a very interesting contribution of outstanding usefulness for future research on global soil biodiversity and functions. The topic is highly relevant to the global environmental context and the need for humanity to secure the production of goods and services from healthy ecosystems and sustainable agrosystems. It is highly strategic as it aims at identifying gaps in our knowledge, not only on a geographical basis, but also on an environmental basis, i.e. identifying blind points in our knowledge in terms of environmental categories that have been under-explored in the scientific research dedicated to soil biodiversity and soil ecological functions. The approach is sound, and there is no doubt that, once published, it will become a reference for justifying more studies on soil biodiversity and functions in specific areas of the world, and a basis to fill the gaps in our knowledge in this research area.

The paper is globally written in a very understandable way. Methods are relevant to the question, even if I have some minor points I would like the author to consider for clarification. The discussion is sound and justified by the results of the meta-analysis in its first two sections (2.1. "Biogeographical biases" and 2.2 "2.2. Ecological blind spots"), then move to a very interesting third section that discuss the main constraints soil ecological research is currently facing and proposes some solution to bypass them (2.3 "Challenges to move beyond blind spots").

Here is a list of minor comments and recommendations:

- Lines 102-103: I am not sure to understand correctly this sentence. Do you mean fungi and bacteria have been the focus of the main taxonomic research efforts dedicated to soil biota? I am not sure this is the case, and soil micro-organisms in general are probably concerned by a very high taxonomic deficit,

even in temperate areas (see for instance the figure with taxonomic deficit by main taxa of soil organisms, and related to the average body-mass in Decaëns et al. 2010).

- Line 131: what do you mean here by "individual sampling site"? This should represent different spatial scales across studies and focal groups of soil organisms or ecosystem functions. In the method section, lines 763-767 no additional information is given regarding this point.

- Line 723-735: I would have expected to see termites specifically included in the keywords. This is a quite important group of soil ecosystem engineer, involved in a number of key ecosystem functions especially in the tropics (but not exclusively), and a number of studies on global patterns of termite biodiversity have been already published (e.g. Eggleton 2000, Eggleton et al. 1994).

- Figure 2: "Values (y-axis) correspond to the percentage of sites per study per class when compared with the global proportional distribution of each class within each variable defining soil ecosystems"... This is not very easy to follow. Could you explain this in a more simple way?

- Figure 2, what are the individual data units in this figure? From the legend I understand that these are "studies", but in this case how can you calculate the % of under/over-representation individually at the study scale? The method section is useless to respond to this question, and this important figure absolutely has to be self-understandable.

- Figure 2, it would probably also be interesting to check the representation of each major biome, and of the different latitudinal bands. This is addressed in Figure S3, but in a different way, and it would certainly be interesting to highlight here the bias in available data towards temperate areas. This could be integrated into Figure 2 or could represent a separate figure integrated to section 2.1.

- Still in Figure 2, I wonder if the categorization of land covers is really relevant. What are "lichens and mosses" representing in a land cover classification? Also, are you sure of this over-representation of urban areas or could this be an artefact of geo-coordinate approximations? Explain the meaning of "mosaic (crops)" and "mosaic (forests)".

- In the text, you refer to Figure 2 to point blind spots (poorly covered environmental categories) and hot spots (environmental categories over-represented in macroecological literature) in soil biodiversity/function knowledge. However I am not sure that you mention in any part of the manuscript the threshold above which you will consider the % of representativeness to be different from zero (representation of an environmental category in macroecological literature proportional to its

representation in the real world). Is this a hundred %, or more, or less? This should be specified, justified and could also be represented in Figure 2 by two horizontal dotted lines (one above and the other one below the zero line) that would help to visualise which boxplots represent a significant trend towards over or under-representation.

- Lines 256-263: I suspect that most of the trends illustrated in Figure 2 (especially with regard to climate coverage) can be explained by a latitudinal gradient, which is detectable in Figure 1b-c, but not explicitly identified in the manuscript (see my previous comment recommending the illustration of the latitudinal gradient and/or main biomes in Figure 2). In tropical areas, many other reasons for a lack of research on soil biodiversity/function can be advocated: lack of available support and infrastructure for soil research, huge taxonomic deficit on tropical biota, which impairs efficient and repeatable soil biodiversity surveys, etc...

- Figure 3: what are represented by the pie charts on the right side of the figures? I do not understand the different percentages indicated in these...

- Fig. S1: in the figure legend "soil mentions" instead of "soi mentions"

- Fig. S2a: Could you specify the number of studies (or individual sampling sites) represented in each map? It seems strange that you have no more data for e.g. collembolan and acari.

References

Decaëns T (2010) Macroecological patterns in soil communities. *Global Ecology and Biogeography*.

Eggleton P (2000) Global patterns of termite diversity. In: *Termites: evolution, sociality, symbioses, ecology*; Springer.

Eggleton P, Williams PH, Gaston KJ (1994) Explaining global termite diversity: productivity or history? *Biodiversity & Conservation*.

Reviewer #3:

Remarks to the Author:

The study by Guerra and collaborators, entitled "Blind spots in global soil biodiversity and ecosystem function research" compiled the location of ~11K soil studies, some characterizing soil biodiversity, some ecosystem functions, and few both. The authors then proceed to map these studies, and to identify the density of studies across areas and categories of soil properties ("blind spots"). Finally, authors suggest the creating of a network of soil research to share soil data and coordinate sampling in areas of poor knowledge.

I enjoyed reading the manuscript, which is well written. I'm not a soil ecologist, and had no prior knowledge on the state-of-art of coverage of soil sampling worldwide.

However, I have a concern regarding an underlying assumption of the study: all regions require the same number (or density) of samples to characterize the biodiversity or ecosystem functions in the region. Although the authors do not state the assumption explicitly, inferring blind spots based on the absolute count (or density) of soil samples assumes that (1) soil biodiversity and ecosystem functions are evenly distributed across space, therefore (2) studies should be evenly distributed. In addition, the assumption is not only limited to regions (area), but also to the categories of environmental properties. For example, the authors claim that soils in areas of low plant diversity (Fig. 2m), extremely cold (Fig. 2j) and extremely dry (Fig. 2k) are poorly studied, relative to the frequency of those categories across the world.

Again, I'm not a soil ecologist, but I this assumption is most likely unrealistic. The number of studies necessary to characterize the biodiversity should be proportional to the local biodiversity, and not evenly distributed across space. In other words, if the goal is to characterize biodiversity, then sampling the soil in areas of lowest plant richness, lowest temperature and lowest precipitation may be a waste of time and resources, as these areas are likely to contribute little to soil biodiversity and ecosystem functions. So, here I assume the exact opposite of what the authors have assumed: regions of highest biodiversity should concentrate most studies, and extreme environments are likely to harbor lowest biodiversity (at least for aboveground terrestrial biodiversity).

The unrealistic assumption of the study emerges from use of relative frequencies (densities, such as number of studies within 1-degree grid cell) of studies as an indication of knowledge. In fact, modern methods to estimate biodiversity assume that one should contrast known biodiversity (e.g. species inventories) against estimated/expected biodiversity (inferred from previous studies). Thus, regions/environments of poor biodiversity should need lower sampling effort than regions/environments of high biodiversity. In other words, is very likely that biodiversity in regions of extremely low temperature, low precipitation, high aridity and low plant richness is best described than other regions, despite the fewer number of samples. So, it is not necessarily true that blind spots are in regions of low number of samples, as the most sampled regions may need even sampling effort to characterize the local soil biodiversity.

Knowledge gaps (i.e. "blind spots", sensu Guerra and collaborators) are better identified using estimates of Completeness/Coverage (e.g.: Anne Chao and Lou Jost 2012. Coverage-based rarefaction and extrapolation: standardizing samples by completeness rather than size. Ecology 93:2533-2547), which use the "novelty" of each new sample to infer how (in)complete the description of biodiversity is. Thus, I would recommend using such completeness estimators to identify the "blind spots" in soil biodiversity, rather than pure density of number of soil samples.

Thiago F. Rangel
Federal University of Goiás, Brazil

Review: “Blind spots in global soil biodiversity and ecosystem function research”

Reviewers' comments:

Reviewer #1 (Remarks to the Author):

This study provides an overview on potential knowledge gaps in soil biodiversity and ecosystem functioning. As the authors acknowledge, the majority of the studies available in the literature merely focus on microbes, and there is an overall lack of research studies dealing with the temporal variability of many other soil organisms across many soil types, land covers, and biomes. However, the manuscript does not provide enough novel information to make a significant progress that could increase this understanding. Overall, I found the study to be very descriptive and lacking more in-depth analyses of the data extracted from the literature. For example, I would have expected to see (from what we have) whether it is possible to find any meaningful patterns in soil biodiversity and ecosystem function at global scales. This would have allowed a more critical and detailed outcome.

Reply: We thank the reviewer for raising the point that we have to better emphasize the novelty of the present work. We completely understand the concern of this reviewer. However, one of the major points that we highlighted in our paper is actually the lack of studies including both soil biodiversity and ecosystem function data coming from the same plots and points in time. This limits any attempt for more in-depth analyses linking soil biodiversity and function at this point. We have now clarified this important point raised by the reviewer and have also worked to further highlight the novelty of our work in lines 122-129, 201-219, and 265-273. What the reviewer asks for is a data collection effort that is beyond the scope of the current manuscript. We, on the other hand, focus on exposing the potential misleading interpretation of recently published global results, given the limited environmental and global coverage of these datasets, our “blind spots”. In the manuscript, we now clearly state that our efforts were limited to understanding how a wide range of relevant environmental characteristics are represented by the different studies/datasets in use in soil macroecology. Our goal was not to replicate the significant effort taken by e.g. van der Hoogen et al., 2019 (Nature) and Phillips et al., 2019 (Science) to describe soil diversity (in all cases these deal with single groups and took years of collective effort from hundreds of soil ecologists). We rather focus on the hidden aspect of these manuscripts, that is, how well they represent global environmental conditions, why the revealed limitations exist, and how we can move beyond them. This new information is missing from most of the included studies and should both reshape and target our efforts to have more representative conclusions for global soil macroecology. We also highlight a clear path to move forward, beyond current limitations.

Like the reviewer, we also share a strong motivation to explore soil BEF relationships and would have liked to test them. Unfortunately, as we point out in the manuscript, this is not possible from the existing soil macroecological literature due to both a limiting number of global sites, very limited environmental representation, and a strong mismatch of biodiversity groups and

functions being examined in each of the papers reviewed. Notably, we - in collaboration with other soil ecologists - recently established a global soil biodiversity and function monitoring network (Soil BON: <https://geobon.org/bons/thematic-bon/soil-bon>) as well as the Global Initiative on Crop Microbiome and Sustainable Agriculture (<https://www.globalsustainableagriculture.org>) that are supposed to deliver the kind of data we and the reviewer would like to have. Nevertheless, we expanded on the constructive comments from the reviewer and provide more detailed answers below.

General comments:

The fact that studies dealing with soil biodiversity and ecosystem functioning at large scales are a relative recent research field, the selection of the studies appears to be strongly biased and not properly justified. For example, due the scanty number of macroecological studies it would have been useful to also include those individual studies focusing at national levels. In addition, using 1-3 studies to make conclusions about metazoans and four soil functions (according to Table 2 in the methods) is not appropriate. In fact, the maps provided in Figs. S4 are misleading because if I am right, the authors only included one study to draw the maps of Collembolans (Fig. S2a/g), nutrient cycling (Fig. S2b/c) and water infiltration (Fig. S2b/d). In addition, also according to Table 2 in the methods section, the same study (e.g., 151 and 153) have been used to draw different maps.

Reply: The scope of the present paper was to analyze how well-published soil macroecological studies on specific taxa and ecosystem functions represent global environmental conditions to guide future synthesis and monitoring efforts. Notably, it is beyond the scope of this study to summarize all available local datasets on different soil taxa and functions. We rather focused on past and current global efforts aiming to investigate the distributions of soil biodiversity and ecosystem functions worldwide. While we continue to support such activities, we underline that the present manuscript was supposed to provide necessary baseline information and to highlight critical gaps in current datasets in an incredibly vibrant field of research. We thus believe that the novel results provided here will be of high interest to many scientists and serve as an important guide to future research efforts to fill these knowledge gaps.

In the abstract, the authors claim that > 11,000 sampling sites were included, but the actual analyses (and therefore the conclusions) are only based on 45 studies. According to the text, the initial number of sites (11,065) consisted of 7,631 sites reporting soil biodiversity data and 3,497 sites with information on soil functions (L131-133). Did all these sampled sites were obtained from just 45 studies (L757-758)? In L168 it is stated in that 63 sites contained both types of information, should not they be presented to the reader in a map? This would have given a more clear idea of where this type of information is concentrated. And, did all the soil biodiversity sites have information on several soil organisms or just specific groups?

Reply: We appreciate the reviewer for the very helpful comment. In response to the comment, we added Fig. S6 that clearly depicts the distribution of the sites with both biodiversity and ecosystem functions and also shows the mismatch between taxa and functions across the sites. Regarding the studies used these are all listed in Table 2, but yes, although now the number of studies and samples have increased the listed information came from 45 studies.

I am not very convinced that data are mostly absent from most tropical areas (L102) when one of the first international initiatives Tropical Soil Biology and Fertility Programme in the 80s was focused on that particular biome. I was also surprised to see that The International Biological Program (IBP), an effort between 1964 and 1974 to coordinate large-scale ecological and environmental studies, was not even mentioned. It is also surprising that one charismatic group of tropical systems, termites, were not included or mentioned in the paper. And what about all the soil atlases produced by FAO, USDA and ESDAC that provide abundant soil data?

Reply: We appreciate this comment and apologise for the confusion. Please, note that it was not our intention to claim that there is no data for tropical soil ecosystems. Our claim is that soil global surveys using comparable methodologies (sampling, sequencing etc.) often lack comprehensive data for these ecosystem types. Briefly, this aspect is even more critical when most global studies focus on “average solutions” (i.e., due to the lack of regionalization of the analysis) to map global distributions, leading to tropical sites to appear as environmental outliers. This means that the regression or other models and analyses built with these datasets are strongly influenced by average (typically temperate) environmental conditions and do not represent tropical conditions, even when they are included (typically with less than 1% of the dataset). We have made an effort to make this very important point more clear in the revised manuscript (lines 179-182).

In addition to these large scales projects, there are currently several databases that continuously archive soil organisms and/or soil data, such as Edaphobase, DriloBase, Collembola database, etc. Therefore, I would have expected that the first effort should be the incorporation of all this information in a study that intends to identify knowledge gaps. In addition, before we care about macroecological gaps, should we not aim for a better knowledge of soil biodiversity when it has been stated that we only know a small fraction (e.g. in the macroecological study by Decäens 2010)?

Reply: The reviewer has a valid and important argument about the potential inclusion of information archived in databases like Edaphobase, DriloBase, Microbiome, and even GBIF. We totally agree that these are fantastic sources of information; however, to the best of our knowledge, most of the data archived in these databases are not used, nor ready to use, by soil macroecological studies due to the lack of standardization, harmonization, and a range of access conditions to the actual data. Because of this, we decided not to actively include these data in our

analyses, but only refer to published studies. However, we appreciate this important comment, and have now added content to the main text highlighting the future importance of these databases (lines 109-116).

We also would like to highlight that our aim was to show that most global conclusions coming from macroecological studies lack global representation (clarified in lines 170-183) and, therefore, not to discredit the conclusions of these studies, but rather to call for more representative datasets and efforts to close these gaps in knowledge. In this context, our focus has to be on published studies and not on potentially available information coming from data repositories that, so far, present limiting conditions for macroecological studies, requiring significant standardization and harmonization before entering a proper macroecological analysis. Basically, this manuscript represents a fitness check of available macroecological studies to show not only the expected global environmental “blind spots” of soil macroecology but also the less expected ones that are driven by current limitations in access to soil biodiversity and ecosystem function FAIR datasets. To make these points clear in the text, we extended our discussion section and included new analyses (see new Fig. 3 and the comments below for more specifics on the latter).

More specific comments:

In several parts of the text there is an overemphasis of the described findings; for example, the use of the term “significant” is misleading when no statistical tests have been applied to the data reported.

Reply: Agreed. We have now modified our manuscript accounting for this issue (e.g., lines 61, 94, 140, 149, 264, 270, 273, 417, 490, 984).

Some kind of description/justification is needed to explain how the class intervals for each of the variables depicted in Fig. 2 were established. Were they based on a literature survey or on a specific criterion? Also in this multi-panel figure, Fig. 2n clearly shows that urban areas are overrepresented, what does this mean in terms of number of studies analysed? How can you be sure that climatic, soil and above-ground variables are the main determinants of edaphic life and not anthropogenic disturbances?

Reply: We would like to thank the reviewer for these valuable comments and guiding questions. Regarding the figure and the classification of the class intervals, this was done by applying the natural breaks classification (*sensu* Jenks), as it is explained in the methods (lines 973-985) and now clarified in the figure caption. Regarding the comment on urban areas over-representation, we actually think that this may be a consequence of two issues. One is the precision of the coordinates together with the precision of the land-use classification. The second is related to the expansion of urban areas and the fact that most samples are taken in close proximity of cities and other human occupied areas. Without the capacity to properly validate the accuracy of the site

locations, we can only speculate regarding this over-representation. We added these potential caveats to the Discussion (lines 305-311).

The same can be said for the other charts, how many studies reported these values? In several of these charts, 20 intervals were established, meaning that 2 studies were represented in each one... Also units for Figs. 2f-n? Vascular plant richness is also included to describe “global soil ecological blind spots” (Fig. 2m), but in several parts of the text it is stated that no links between above- and below-ground biodiversity has been found, so which are the reasons behind using this metric as potential influential factor explaining knowledge gaps in soil biodiversity and ecosystem functioning?

Reply: We apologize for this lapse on our part. We have now included the units of all variables in the figure caption. Regarding the inclusion of plant diversity, although most macroecological studies point in the direction of a mismatch between above- and belowground diversity, there are still important dependencies between above- and belowground diversity, such as experimental evidence for detrimental consequences of local plant diversity loss for soil biodiversity and functions. These dependencies can be functional and, in the case of some groups (e.g., like fungi) can also affect the overall distribution of soil organisms. We have included this clarification in the text.

I do agree that lack of standardized protocols hinders our understanding, so how was this solved in this study when compiling the data from the different studies? For example, measuring C content by loss of ignition, dichromate-oxidation method, etc. could render different results, as well as different methods to extract soil biota. And what about the use of different soil classifications when plotting soil types?

Reply: This is a great point. Regrettably as this reviewer pointed out, the current lack of standardized protocols did not allow for a better approach. However, future studies should focus on specific protocols to solve this issue (lines 359-362, 392-394, 490-494).

A justification is also needed to explain the reasons behind combining just three variables when trying to assess which environmental characteristics are used in macroecological studies (Fig. 3). Since that according to Fig. 2 and the explanations given in the text, soil carbon, land cover are biased, should it not be better to illustrate the combination of variables for which we have good coverage? Why was not soil type included? This information is usually available in many published soil atlases...

Reply: To address this important comment, we have now changed our methodological approach to have a more statistically based description of environmental coverage (as described in lines 998-1007). Given the factorial nature of the analysis previously in Fig. 3, we were not able to

include soil type (30 broad soil types) as it would increase dramatically the number of possible combinations and further increase the amount of blind spots. We do not claim that every single combination should/needs to be covered across the globe, and adding this very diverse variable would result in an overestimation of the environmental blind spots.

I like the overview of the main obstacles that soil ecologists face when trying to produce/complete soil surveys and that has been summarised in Table 1. In my opinion, scientific challenges are also important. Despite the authors trying to convince the reader that data sharing and making data available to be re-used again could be the solution to decrease the number of knowledge gaps, I feel that this is only one side of the story. Scientists from around the world need to publish their findings to progress in their careers and/or to obtain research funding. Managing large data sets, having access to global databases on climatic and soil information is often a privilege for rich research teams (in terms of bibliographic subscriptions, computing power and technical knowledge, and graphical software). The text in L365-371 gives the impression that “data mobilisation” should be a priority and that by embracing open access and connecting the global soil community, we could fulfil all those missing knowledge gaps. However, due the current taxonomic deficit for soil biota, the priority should be to have more national (and local) surveys to increase the soil biodiversity information across a large number of sites and to a deeper taxonomical level. Global research projects (L359) would be ideal but unrealistic in many practical terms.

Reply: We completely agree with the reviewer. We changed the text to clarify these ideas and added the following text inspired in the reviewer’s comments in lines 424-434.

Should not the authors acknowledge the authors who helped when obtained the locations of the sampled sites? This is an interesting study that could attract many potential readers if it could provide a clearer message and quantitative analyses (including statistical tests and not just maps) of the available data.

Reply: We thank the reviewer for the constructive comments. We also agree with the recognition of the authors contributing data to this manuscript suggested by the reviewer. This was now included in the Acknowledgements.

Reviewer #2 (Remarks to the Author):

The manuscript from Guerra et al. “Blind spots in global soil biodiversity and ecosystem function research” represents to me a very interesting contribution of outstanding usefulness for future research on global soil biodiversity and functions. The topic is highly relevant to the global environmental context and the need for humanity to secure the production of goods and

services from healthy ecosystems and sustainable agrosystems. It is highly strategic as it aims at identifying gaps in our knowledge, not only on a geographical basis, but also on an environmental basis, i.e. identifying blind points in our knowledge in terms of environmental categories that have been under-explored in the scientific research dedicated to soil biodiversity and soil ecological functions. The approach is sound, and there is no doubt that, once published, it will become a reference for justifying more studies on soil biodiversity and functions in specific areas of the world, and a basis to fill the gaps in our knowledge in this research area. The paper is globally written in a very understandable way. Methods are relevant to the question, even if I have some minor points I would like the author to consider for clarification. The discussion is sound and justified by the results of the meta-analysis in its first two sections (2.1. “Biogeographical biases” and 2.2 “2.2. Ecological blind spots”), then move to a very interesting third section that discuss the main constraints soil ecological research is currently facing and proposes some solution to bypass them (2.3 “Challenges to move beyond blind spots”).

Reply: We appreciate the reviewer for the very constructive and supportive comments! We addressed all of the comments provided below.

Here is a list of minor comments and recommendations:

- Lines 102-103: I am not sure to understand correctly this sentence. Do you mean fungi and bacteria have been the focus of the main taxonomic research efforts dedicated to soil biota? I am not sure this is the case, and soil micro-organisms in general are probably concerned by a very high taxonomic deficit, even in temperate areas (see for instance the figure with taxonomic deficit by main taxa of soil organisms, and related to the average body-mass in Decaëns et al. 2010).

Reply: We appreciate the reviewer’s comments and agree that clarification was needed. We added the following sentence to the text: “Also, even for the most representative microbial taxa (i.e. Bacteria and fungi), there are strong concerns regarding the current taxonomic depth, even in very well represented regions.”

- Line 131: what do you mean here by “individual sampling site”? This should represent different spatial scales across studies and focal groups of soil organisms or ecosystem functions. In the method section, lines 763-767 no additional information is given regarding this point.

Reply: True. Here we refer to locations, basically unique individual X,Y pairs of coordinates. In the methods, we now included the text: “In the manuscript, sampling sites refer to unique X,Y pairs of coordinates that represent individual locations.”

- Line 723-735: I would have expected to see termites specifically included in the keywords. This is a quite important group of soil ecosystem engineer, involved in a number of key ecosystem functions especially in the tropics (but not exclusively), and a number of studies on global patterns of termite biodiversity have been already published (e.g. Eggleton 2000, Eggleton et al. 1994).

Reply: We agree with the reviewer. To address this point, we ran a new literature search with termites as a search term and all the same search terms as before. This added 25 new studies to our original pool of studies, but none of them qualified to be included in our analysis according to our original criteria, as defined in the Methods section (5.1. Literature selection and data processing). You can find the 25 papers here: http://tiny.cc/blindspots_search according to the following search criteria: (Global* OR Continental OR latitud*) AND (soil* OR belowground) AND (*function* OR *diversity OR organism* OR biota OR animal* OR invert* OR fauna*) AND distribution AND termite*. Importantly, we do not claim to provide a comprehensive overview of the literature but rather a sound and standardized representation of the bulk of the current soil macroecological literature.

- Figure 2: “Values (y-axis) correspond to the percentage of sites per study per class when compared with the global proportional distribution of each class within each variable defining soil ecosystems”... This is not very easy to follow. Could you explain this in a more simple way?

Reply: We agree with the reviewer. We rephrased the caption to clarify it. It now reads: “Values (y-axis) correspond to the percentage of sites per study when compared with the global proportional distribution (e.g., a value of 20% means that a given study over-represents a given environmental variable by 20%, when compared to the global distribution of that same variable).”

- Figure 2, what are the individual data units in this figure? From the legend I understand that these are “studies”, but in this case how can you calculate the % of under/over-representation individually at the study scale? The method section is useless to respond to this question, and this important figure absolutely has to be self-understandable.

Reply: Following the previous comment, we changed the figure caption to capture the reviewer’s concerns. We added the following statement: “Each barplot represents the proportional number of sampling sites covering a particular class when compared to the global distribution.”

- Figure 2, it would probably also be interesting to check the representation of each major biome, and of the different latitudinal bands. This is addressed in Figure S3, but in a different way, and it would certainly be interesting to highlight here the bias in available data towards temperate

areas. This could be integrated into Figure 2 or could represent a separate figure integrated to section 2.1.

Reply: We have now included Fig. S7 that describes the latitudinal differences in terms of sample distribution and expanded on this topic on the text (lines 210-214)

- Still in Figure 2, I wonder if the categorization of land covers is really relevant. What are “lichens and mosses” representing in a land cover classification? Also, are you sure of this over-representation of urban areas or could this be an artefact of geo-coordinate approximations? Explain the meaning of “mosaic (crops)” and “mosaic (forests)”.

Reply: Regarding the classification of land-use types, we followed an available classification by the European Space Agency (see Fig. S4), which also includes lichens and mosses. We agree that some classes may not be as intuitive as others. Mosaic (crops) represent small-scale landscapes dominated by crops, while mosaic (forests) represent small-scale landscapes dominated by forests. This clarification was added to the caption. However, we still used this classification, because it is already publicly available and it is largely used in the current literature and the supporting land cover data infrastructures (e.g., <http://maps.elie.ucl.ac.be/CCI/viewer/index.php>). Regarding the comment on urban areas over-representation, we actually think that this may be a consequence of two issues. One is the precision of the coordinates together with the precision of the land-use classification. The second is related to the expansion of urban areas and the fact that most samples are taken in close proximity of cities and other human occupied areas. Without the capacity to properly validate the accuracy of the site locations, we can only speculate regarding this over-representation. We added these potential caveats to the Discussion (lines 305-311).

- In the text, you refer to Figure 2 to point blind spots (poorly covered environmental categories) and hot spots (environmental categories over-represented in macroecological literature) in soil biodiversity/function knowledge. However I am not sure that you mention in any part of the manuscript the threshold above which you will consider the % of representativeness to be different from zero (representation of an environmental category in macroecological literature proportional to its representation in the real world). Is this a hundred %, or more, or less? This should be specified, justified and could also be represented in Figure 2 by two horizontal dotted lines (one above and the other one below the zero line) that would help to visualise which boxplots represent a significant trend towards over or under-representation.

Reply: Given that we are looking at the overall distribution of all studies, we did not want to introduce such a threshold in our analysis. Although we see the point made by the reviewer, this threshold would be necessarily arbitrary and after discussion we opted not to include it. Furthermore, in the new Fig. 3 that is based in statistical thresholds to identify coverage outliers, this is more clearly identified without arbitrary classifications.

- Lines 256-263: *I suspect that most of the trends illustrated in Figure 2 (especially with regard to climate coverage) can be explained by a latitudinal gradient, which is detectable in Figure 1b-c, but not explicitly identified in the manuscript (see my previous comment recommending the illustration of the latitudinal gradient and/or main biomes in Figure 2). In tropical areas, many other reasons for a lack of research on soil biodiversity/function can be advocated: lack of available support and infrastructure for soil research, huge taxonomic deficit on tropical biota, which impairs efficient and repeatable soil biodiversity surveys, etc...*

Reply: We have now included Fig. S7 that describes the latitudinal differences in terms of sample distribution.

- *Figure 3: what are represented by the pie charts on the right side of the figures? I do not understand the different percentages indicated in these...*

Reply: We now changed this figure following a reviewer's recommendation.

- *Fig. S1: in the figure legend "soil mentions" instead of "soi mentions"*

Reply: This typo was corrected. Thank you for spotting it.

- *Fig. S2a: Could you specify the number of studies (or individual sampling sites) represented in each map? It seems strange that you have no more data for e.g. collembolan and acari.*

Reply: Very good suggestion. We added the n values to the figure. The lower representation of some groups comes from both the low amount of studies found and from the low amount of studies that fit our inclusion criteria (please note that we did not include any local studies; this was beyond the scope of this paper).

Reviewer #3 (Remarks to the Author):

The study by Guerra and collaborators, entitled "Blind spots in global soil biodiversity and ecosystem function research" compiled the location of ~11K soil studies, some characterizing soil biodiversity, some ecosystem functions, and few both. The authors then proceed to map these studies, and to identify the density of studies across areas and categories of soil properties ("blind spots"). Finally, authors suggest the creating of a network of soil research to share soil data and coordinate sampling in areas of poor knowledge.

I enjoyed reading the manuscript, which is well written. I'm not a soil ecologist, and had no prior knowledge on the state-of-art of coverage of soil sampling worldwide.

Reply: We thank the reviewer for the positive and constructive comments and addressed all of them below. We very much appreciate the feedback that our work is well written and provides important insights for non-specialized readers.

However, I have a concern regarding an underlying assumption of the study: all regions require the same number (or density) of samples to characterize the biodiversity or ecosystem functions in the region. Although the authors do not state the assumption explicitly, inferring blind spots based on the absolute count (or density) of soil samples assumes that (1) soil biodiversity and ecosystem functions are evenly distributed across space, therefore (2) studies should be evenly distributed. In addition, the assumption is not only limited to regions (area), but also to the categories of environmental properties. For example, the authors claim that soils in areas of low plant diversity (Fig. 2m), extremely cold (Fig. 2j) and extremely dry (Fig. 2k) are poorly studied, relative to the frequency of those categories across the world.

Reply: The analysis of the reviewer is in general correct. There is not a direct assumption that all characteristics have to be sampled equally, although the data represented in Figure 2 and Figure 3 tends to give this impression. To start, what we really want to illustrate (Figure 2) is that many global environments are completely underrepresented while others appear to be overrepresented and that these features are systematic across soil macroecological studies, suggesting an overrepresentation of certain regions of the globe (e.g., Europe). Second, we have changed the analytical approach in Figure 3 to avoid these confusions (please see our comments below for further explanation on this point) as we go beyond specific variables and try to understand the spatial coverage of current environmental gradients. To tackle this issue raised by the reviewer, we included recent references illustrating the global distribution of different soil organisms to further argue our focus on environmental coverage, and clarified the aspects related to our assumptions. We hope that our changes satisfy the points raised by the reviewer.

Again, I'm not a soil ecologist, but I this assumption is most likely unrealistic. The number of studies necessary to characterize the biodiversity should be proportional to the local biodiversity, and not evenly distributed across space. In other words, if the goal is to characterize biodiversity, then sampling the soil in areas of lowest plant richness, lowest temperature and lowest precipitation may be a waste of time and resources, as these areas are likely to contribute little to soil biodiversity and ecosystem functions. So, here I assume the exact opposite of what the authors have assumed: regions of highest biodiversity should concentrate most studies, and extreme environments are likely to harbor lowest biodiversity (at least for aboveground terrestrial biodiversity).

The unrealistic assumption of the study emerges from use of relative frequencies (densities, such as number of studies within 1-degree grid cell) of studies as an indication of knowledge. In fact, modern methods to estimate biodiversity assume that one should contrast known biodiversity (e.g. species inventories) against estimated/expected biodiversity (inferred from previous

studies). Thus, regions/environments of poor biodiversity should need lower sampling effort than regions/environments of high biodiversity. In other words, is very likely that biodiversity in regions of extremely low temperature, low precipitation, high aridity and low plant richness is best described than other regions, despite the fewer number of samples. So, it is not necessarily true that blind spots are in regions of low number of samples, as the most sampled regions may need even sampling effort to characterize the local soil biodiversity.

Knowledge gaps (i.e. “blind spots”, sensu Guerra and collaborators) are better identified using estimates of Completeness/Coverage (e.g.: Anne Chao and Lou Jost 2012. Coverage-based rarefaction and extrapolation: standardizing samples by completeness rather than size. Ecology 93:2533-2547), which use the “novelty” of each new sample to infer how (in)complete the description of biodiversity is. Thus, I would recommend using such completeness estimators to identify the “blind spots” in soil biodiversity, rather than pure density of number of soil samples.

Reply: We agree in general terms with the reviewer and appreciate these comments. Before responding, we would like to highlight that we do not have access to the actual biodiversity/function data (apart from some publicly available references) limiting the use of diversity driven approaches. We are approaching this from the assumption that current global soil biodiversity hotspots are either poorly characterized (due to data constraints) or fail to account for the diversity of environmental conditions present in the globe. Therefore, we argue that more environmental conditions should be the focus of soil ecologist and that, at this stage, these conditions should be targeted equally, without an overwhelming bias towards specific regions of the world as typically diverse areas (e.g., tropical areas) are often poorly represented in macroecological studies. Once this is done, we agree with the reviewer that hotspots could be targeted, but unfortunately this would probably undermine even more the taxonomic depth that even temperate (over represented) regions already have. We also agree that regions/environments of poor biodiversity could need lower sampling effort than regions/environments of high biodiversity. However, we would like to stress that (1) we still lack studies identifying those regions with low and high diversity, and (2) wide gradients of locations including both low and high diversity and functioning are needed to test for biodiversity-function relationship hypotheses. All these aspects were clarified in lines 1002-1012. In addition, some important research questions about the spatial scaling of biodiversity, such as alpha- versus beta-diversity distribution (e.g., Phillips et al. 2019 Science) will need additional sampling efforts across many regions around the world, particularly because soil organisms may span a massive gradient in spatial-scaling dependencies (Thakur et al. 2019 Biol Rev). We added this point to the revised manuscript. Our argument is that, given the lack of knowledge (Cameron et al., 2018, NEE), soil macroecological studies need to focus on having a proper representation of the global environmental spectrum. Finally, although we agree that greater understanding of some of these issues regarding representativeness could be achieved by methods that include biodiversity metrics, here we did not have access to the raw data needed to conduct them. To overcome these issues, we included a new analysis (new Fig. 3) that focuses on the statistical analysis of the

coverage of environmental conditions with the aim to provide similar information to that described by the reviewer in terms of identifying the statistical environmental outliers of the different studies.

Reviewers' Comments:

Reviewer #1:

Remarks to the Author:

The authors have done a good job in addressing some of the comments I brought up in the previous version, but my main concern still remains and I still feel that the paper adds very little scientific progress to what is currently known. While the text and the presentation of the results are clearer in the current version, I still find it highly descriptive, with too much text (I cannot see a clear separation between sections 2.1 and 2.2, and between 2.3 and 2.4). Surely, the key messages can be delivered in 2-3 pages and 2-3 figures.

In my previous report, I suggested to perform more in-depth analyses of the data extracted from the literature. The authors argue that this is beyond the scope of the ms, but, in my opinion, this decreases its value. I am "only" seeing a great effort in collating study records and the description of what is known or not known using a vote-counting system which does not grant such lengthy paper. I cannot see why the authors could not have used statistical tools such as meta-analysis to draw some meaningful conclusions on the available literature: e.g. Can we quantitative link more collembolans to more soil respiration? Are drylands losing more carbon than forest due to soil biodiversity differences? Are the predictive factors we are measuring now (soil, organisms and environmental variables) informative enough and can we trust the knowledge already gathered? Which ecosystems across the globe are maintaining multi-functionality and which ones are under risk of losing it? Again, I apologise for being so critical but as a potential reader, I would have expected a more meaningful outcome from such a high impact journal.

The cited similar work by Decaens (2010) actually contains these kind of analyses I am suggesting. The figures in this paper included analyses of species richness, latitudinal gradients, ecological relationships, etc., in addition to knowledge deficit (also in terms of number of papers dealing with the topic). Sorry to be so critical and please do not get me wrong, I do see enough merits in this study to be published in a high impact journal but perhaps on a more specific ecology journal?

My overall conclusion after reading the re-submitted version still remains and although this is an interesting study that could attract many potential readers, it requires quantitative analyses of the available data to provide a more illuminating message.

In addition, some other comments have not been satisfactorily addressed: From their responses, I got the confirmation that despite the claim in the abstract that > 17,000 sampling sites were included, only a much smaller number (45 studies) have been used in the spatial analyses. It is also confirmed that in the case of some maps, less than 3 studies were used (see my previous comments about Fig. S2). Therefore, the abstract promises much more than the rest of the text delivers.

The response to my criticism on the lack of standardized protocols when compiling the data from the different studies has been addressed as a recommendation for future research but not by indicating how this issue has been tackled in this paper.

Other specific comments:

L57: "T"o contribute?

L60: Soil biodiversity or specific taxa?

L61: The number of studies used for the spatial analyses should be indicated here.

L74 (and throughout the text): Oligochaeta include groups that belong to macro- and meso-fauna. I think the authors are referring to earthworms.

L221-242: These two paragraphs appear to be misplaced in this section because they refer to biogeographical aspects and repeat the same statement derived from the previous section, i.e. biases towards temperate regions.

L149: What do you mean with "organism biomass"? total soil fauna biomass? Specific taxa biomass? How were these data analysed, e.g. if you have studies dealing with a specific group and others dealing with several groups of soil organisms?

L155-158: How many records did you collect for the missing groups mentioned in the methods (e.g. termites, earthworms, etc.)?

L172: Fig. 2e should be accompanied by a column chart showing the distribution of biomass data per taxonomic group (equivalent to Fig. 2b).

L190: Please complete this sentence and indicate which other soil functions have been included in the remaining studies.

L192 and 196: They both repeat the same information.

L255: pH > 7.33 and silt content < 19% are extreme ranges?

L268: I do not understand the point you are trying to make here... How did you estimate the "50.6% of the global climate"?

L271: Sorry, I am not able to grasp the meaning of this graph. According to the legend, it should represent "ecological blind spots", that is knowledge gaps in

the ecological relationships between soil biodiversity and their environment. However, the graph shows, side by side, how well represented soil biodiversity and soil functions are.

L277: A symbol missing here (mean annual temperature)?

L310-311: Really? According to the methods and the supplementary material, only three properties (organic carbon content, sand content and pH) were used in the analyses.

L309-340: Are all these paragraphs and Fig. 3 not describing biogeographical biases (section 2.1)?

L329: I like this graph which is very much along my suggestion of producing some statistically sound results. In the legend of this figure, do you mean "The extent to which mean soil biodiversity and ecosystem functions are...."? Please also indicate the meaning of the square slices.

L358-367: I think this paragraph should be integrated in the next section.

L979: Table 3 is based on 7 studies only?

Reviewer #2:

Remarks to the Author:

I had a careful reading of the manuscript "Blind spots in global soil biodiversity and ecosystem function research" from Gerra et al., and was fully satisfied to see that they take into account all the comments I made on the first version of their work. The few and minor concerns I had on the initial version have all been corrected, or the authors provided convincing arguments to justify their position. I therefore have no more comments to add on the new version. I want to acknowledge the authors for the time they have spent in carrying out this outstanding synthesis, and for the efforts they produced at improving their manuscript.

Reply to Reviewers:

Reviewer #1 (Remarks to the Author):

The authors have done a good job in addressing some of the comments I brought up in the previous version, but my main concern still remains and I still feel that the paper adds very little scientific progress to what is currently known. While the text and the presentation of the results are clearer in the current version, I still find it highly descriptive, with too much text (I cannot see a clear separation between sections 2.1 and 2.2, and between 2.3 and 2.4). Surely, the key messages can be delivered in 2-3 pages and 2-3 figures.

In my previous report, I suggested to perform more in-depth analyses of the data extracted from the literature. The authors argue that this is beyond the scope of the ms, but, in my opinion, this decreases its value. I am “only” seeing a great effort in collating study records and the description of what is known or not known using a vote-counting system which does not grant such lengthy paper. I cannot see why the authors could not have used statistical tools such as meta-analysis to draw some meaningful conclusions on the available literature: e.g. Can we quantitative link more collembolans to more soil respiration? Are drylands losing more carbon than forest due to soil biodiversity differences? Are the predictive factors we are measuring now (soil, organisms and environmental variables) informative enough and can we trust the knowledge already gathered? Which ecosystems across the globe are maintaining multi-functionality and which ones are under risk of losing it? Again,

I apologise for being so critical but as a potential reader, I would have expected a more meaningful outcome from such a high impact journal.

The cited similar work by Decäens (2010) actually contains these kind of analyses I am suggesting. The figures in this paper included analyses of species richness, latitudinal gradients, ecological relationships, etc., in addition to knowledge deficit (also in terms of number of papers dealing with the topic). Sorry to be so critical and please do not get me wrong, I do see enough merits in this study to be published in a high impact journal but perhaps on a more specific ecology journal?

My overall conclusion after reading the re-submitted version still remains and although this is an interesting study that could attract many potential readers, it requires quantitative analyses of the available data to provide a more illuminating message.

Reply: We thank the reviewer for the comments provided. As the title of our paper suggests, we are focusing on the unknowns (or better the topics that cannot be answered) rather than the potential links. Our novel contribution is therefore that many of the conclusions that we are getting from current studies (despite their enormous value) are limited in scope and can condition future research due to their narrow environmental representation. In fact, most of these studies have been published in leading journals like Nature, Science, & Nature Communications, which is why this broad audience will also be interested in our present findings. Using the example suggested by the reviewer (i.e., the potential quantitative link more collembolans to more soil respiration), our study suggests that such an analysis cannot be done at the global scale with currently available datasets, at least not in a quantitative way due to the

lack of matching ecosystem function (respiration) and diversity (Collembola) data. Generalizing from severely environmentally skewed data can have important consequences, and results should be treated carefully. In our final revision, we checked again that the respective messages are conveyed in a clear way.

In addition, some other comments have not been satisfactorily addressed:

From their responses, I got the confirmation that despite the claim in the abstract that > 17,000 sampling sites were included, only a much smaller number (45 studies) have been used in the spatial analyses. It is also confirmed that in the case of some maps, less than 3 studies were used (see my previous comments about Fig. S2). Therefore, the abstract promises much more than the rest of the text delivers. The response to my criticism on the lack of standardized protocols when compiling the data from the different studies has been addressed as a recommendation for future research but not by indicating how this issue has been tackled in this paper.

Reply: Thank you for allowing us to clarify this point. As stated in the revised manuscript, we did not make any valuation regarding the biodiversity and/or function values reported by the different studies. We also did not have access to any of the actual values reflecting soil biodiversity or functions, only to the locations of their sampling sites. Because of this, we did not have to handle the different methods nor any standardization across studies since what we wanted to express was the absence (or not) of information about soil biodiversity or function. Regarding the point raised by the reviewer on the number of sites, the 17,180 sampling sites result from the stacking of the different datasets used by each of the 45 studies included in this analysis. Some studies (e.g., Phillips et al., 2019) have >3000 sampling sites and others just a few. To this point, as written in the text, what we actually found was a very small overlap between studies (regarding their sampling sites, i.e., the specific environmental conditions being reported upon). This further limits their integration and, as pointed by the reviewer, we raised this as a recommendation for future research.

Other specific comments:

L57: "T"o contribute?

Reply: This was corrected accordingly.

L60: Soil biodiversity or specific taxa?

Reply: This was corrected accordingly.

L61: The number of studies used for the spatial analyses should be indicated here.

Reply: The abstract was edited according to the Editor's suggestion. This information is already explicit in many other parts of the manuscript, including figure captions and in the methods.

L74 (and throughout the text): Oligochaeta include groups that belong to macro- and meso-fauna. I think the authors are referring to earthworms.

Reply: Thank you for the comment. According to our list of studies, we are not only referring to earthworms (macrofauna) but also enchytraeids (mesofauna), and therefore we would like to represent this group in a more general way.

L221-242: These two paragraphs appear to be misplaced in this section because they refer to biogeographical aspects and repeat the same statement derived from the previous section, i.e. biases towards temperate regions.

Reply: We appreciate the comment from the reviewer. Nevertheless, in our view it is different to describe a bias towards something (e.g., describing a bias towards Europe in terms of sampling locations) and the absence of information regarding a particular topic (e.g., what is reported in Fig. 3). This is also referring to the comment on lines L309-340. In our revised manuscript, we made sure to use clear terminology to avoid any further confusion.

L149: What do you mean with “organism biomass”? total soil fauna biomass? Specific taxa biomass? How were these data analysed, e.g. if you have studies dealing with a specific group and others dealing with several groups of soil organisms?

Reply: To address the reviewer’s questions, we edited the text to clarify this point. It now reads: “...referring to organism biomass [N=977] (e.g. microbial and faunal biomass) as an important link between...”

L155-158: How many records did you collect for the missing groups mentioned in the methods (e.g. termites, earthworms, etc.)?

Reply: We thank the reviewer for the comment, but this is not clear to us. Earthworms are included in our analysis (as the reviewer points out in comment L74). Regarding termites, they were included in our search, but did not return any results consistent with our criteria. We also did a bulk search for all terms (as described in the methods) and did not do separate searches for specific taxa or functions.

L172: Fig. 2e should be accompanied by a column chart showing the distribution of biomass data per taxonomic group (equivalent to Fig. 2b).

Reply: We think the reviewer is referring to figure 1 and not figure 2. In this case, since our search does not allow us to separate between microbial groups, we are not able to build this distinction across all groups evaluated. We agree with the reviewer though that such information would be urgently needed in future assessments. The recent call by Smith and colleagues (2019, <http://www.soil-organisms.org/index.php/SO/article/view/112>) suggests that such work is in preparation.

L190: Please complete this sentence and indicate which other soil functions have been included in the remaining studies.

Reply: To address this comment, we included the reference to the figure where this information is clearly described. It now reads: “In fact, from the five functions assessed here, there is a clear

concentration of studies on soil respiration, accounting for 78.8% [N=2,616] of all function records (Fig. 1d; see Supplementary Figure 3 for more detail).”

L192 and 196: They both repeat the same information.

Reply: We have now corrected this repetition and eliminated the first reference to the topic.

L255: pH > 7.33 and silt content < 19% are extreme ranges?

Reply: We thank the reviewer for the comment. According to the datasets used (Table 3), both pH>7.33 and silt<19% correspond to the top (or bottom in the case of silt) 25% of conditions present on Earth.

L268: I do not understand the point you are trying to make here... How did you estimate the “50.6% of the global climate”?

Reply: Thank you for giving us the opportunity to clarify this point. It now reads: “This issue is further exacerbated when looking at specific climate combinations (Fig. 3c), where 59.6% of the global climate conditions are not covered by any of the studies considered.”

L271: Sorry, I am not able to grasp the meaning of this graph. According to the legend, it should represent “ecological blind spots”, that is knowledge gaps in the ecological relationships between soil biodiversity and their environment. However, the graph shows, side by side, how well represented soil biodiversity and soil functions are.

Reply: We appreciate the comment from the reviewer. From the caption of the figure, “a value of 20% means that a given study over-represents a given environmental variable by 20%, when compared to the global distribution of that same variable”. Therefore, the blindspots correspond here to the characteristics that are underrepresented in the studies included in our manuscript.

L277: A symbol missing here (mean annual temperature)?

Reply: As far as we can see, all symbols are present. The text reads: “mean annual temperature (°C)”.

L310-311: Really? According to the methods and the supplementary material, only three properties (organic carbon content, sand content and pH) were used in the analyses.

Reply: The reviewer is correct: in the supplementary figure, only organic carbon content, sand content, and pH are represented. The exclusion of the other two soil related variables (silt and clay content) was mostly due to the correlation with sand content, although if included the results would be similar to the ones represented in the supplements.

L309-340: Are all these paragraphs and Fig. 3 not describing biogeographical biases (section 2.1)?

Reply: In our view, it is different to describe a bias towards something (e.g., describing a bias towards Europe in terms of sampling locations) and the absence of information regarding a particular topic (e.g., what is reported in Fig. 3).

L329: I like this graph which is very much along my suggestion of producing some statistically sound results. In the legend of this figure, do you mean “The extent to which mean soil biodiversity and ecosystem functions are....”? Please also indicate the meaning of the square slices.

Reply: We thank the reviewer for the positive feedback. According to the question, we revised the caption. It now reads: “Results show that most studies have, on average, a coverage below 50% of all the regions in the world, with the exception of Central and west Europe (f) and Caribbean (for both biodiversity and function), Central and North-East Asia, and North and South America (for ecosystem functions). a-f correspond to zooms on specific areas of the globe.”

L358-367: I think this paragraph should be integrated in the next section.

Reply: We changed this accordingly.

L979: Table 3 is based on 7 studies only?

Reply: Table 3 lists all (15) variables used to describe the global environmental scope. These variables were published in seven different studies and have contributed to hundreds of other studies.

Reviewer #2 (Remarks to the Author):

I had a careful reading of the manuscript “Blind spots in global soil biodiversity and ecosystem function research” from Gerra et al., and was fully satisfied to see that they take into account all the comments I made on the first version of their work. The few and minor concerns I had on the initial version have all been corrected, or the authors provided convincing arguments to justify their position. I therefore have no more comments to add on the new version. I want to acknowledge the authors for the time they have spent in carrying out this outstanding synthesis, and for the efforts they produced at improving their manuscript.

Reply: We appreciate the comments from the reviewer and the positive feedback.